# Genomic analysis of global *Plasmodium vivax* populations reveals insights into the evolution of drug resistance

Gabrielle C. Ngwana-Joseph [1], Jody E. Phelan [1], Emilia Manko [1], Jamille G. Dombrowski[1,2], Simone da Silva Santos[3], Martha Suarez-Mutis[3], Gabriel Vélez-Tobón [4], Alberto Tobón Castaño [4], Ricardo Luiz Dantas Machado[5], Claudio R. F. Marinho[2], Debbie Nolder[6], François Nosten [7,8], Colin J. Sutherland [1,6], Susana Campino [1] ✉ & Taane G. Clark [1,9] ✉

Increasing reports of chloroquine resistance (CQR) in *Plasmodium vivax* endemic regions have led to several countries, including Indonesia, to adopt dihydroarteminsin-piperaquine instead. However, the molecular drivers of CQR remain unclear. Using a genome-wide approach, we perform a genomic analysis of 1534 *P. vivax* isolates across 29 endemic countries, detailing population structure, patterns of relatedness, selection, and resistance profiling, providing insights into potential drivers of CQR. Selective sweeps in a locus proximal to *pvmdr1*, a putative marker for CQR, along with transcriptional regulation genes, distinguish isolates from Indonesia from those in regions where chloroquine remains highly effective. In 106 isolates from Indonesian Papua, the epicentre of CQR, we observe an increasing prevalence of novel SNPs in the candidate resistance gene *pvmrp1* since the introduction of dihydroartemisinin-piperaquine. Overall, we provide novel markers for resistance surveillance, supported by evidence of regions under recent directional selection and temporal analysis in this continually evolving parasite.

*Plasmodium vivax* is the most geographically widespread human malaria parasite and the leading cause of malaria outside of sub-Saharan Africa. In 2022, there were 6.9 million cases across 49 endemic countries across Central and South America, East Africa, Asia, and Oceania[1]. Intensive efforts to combat the deadlier *Plasmodium falciparum*, particularly in areas that are co-endemic with *P. vivax*, has distributed resources away from *P. vivax* control programs, leading to

its emergence as the dominant species, particularly in the Greater Mekong Subregion[1]. The absence of a long term continuous in vitro culture system[2] has meant that our understanding of the parasite's life cycle, transmission, and biology has been limited.

Chloroquine, in combination with primaquine, is the front-line treatment for the radical cure of *P. vivax* malaria in most endemic countries. First documented in the late 1980s, chloroquine resistance

[1]Department of Infection Biology, Faculty of Infectious and Tropical Diseases, London School of Hygiene and Tropical Medicine, London, UK. [2]Department of Parasitology, Institute of Biomedical Sciences, University of São Paulo, São Paulo, Brazil. [3]Oswaldo Cruz Foundation – Fiocruz, Rio de Janeiro, Brazil. [4]Grupo Malaria, Facultad de Medicina, Universidad de Antioquia, Antioquia, Colombia. [5]Centro de Investigação de Microrganismos – CIM, Departamento de Microbiologia e Parasitologia, Universidade Federal Fluminense, Niterói, Brazil. [6]UK Health Security Agency, Malaria Reference Laboratory, London School of Hygiene and Tropical Medicine, London, UK. [7]Centre for Tropical Medicine and Global Health, Nuffield Department of Medicine, University of Oxford, Oxford, UK. [8]Shoklo Malaria Research Unit, Mahidol–Oxford Tropical Medicine Research Unit, Faculty of Tropical Medicine, Mahidol University, Mae Sot, Thailand. [9]Faculty of Epidemiology and Population Health, London School of Hygiene and Tropical Medicine, London, UK. ✉e-mail: Susana.Campino@lshtm.ac.uk; Taane.Clark@lshtm.ac.uk

(CQR) in *P. vivax*, also known as chloroquine treatment failure, is characterised by the World Health Organization (WHO) as the persistence of parasitaemia on day 28 following treatment, despite a blood concentration of chloroquine-desethylchloroquine at or above 100 ng/mL. CQR has consequently led to the adoption of artemisinin-based combination therapies to replace chloroquine in several countries, including dihydroartemisinin-piperaquine in Indonesia, artesunate-mefloquine in Cambodia, and artemether-lumefantrine in Papua New Guinea (PNG)[3,4]. Indonesian Papua and PNG have been the epicentre of high-grade, or category 1 CQR, defined as >10% recurrences by day 28 of treatment[5]. Over the past two decades, increasing reports of CQR in *P. vivax* beyond these countries have been characterised by parasite persistence 28 days post-treatment, spurring research into the molecular determinants of CQR[5–9].

Evidence regarding the molecular drivers of CQR in *P. vivax* is both weak and conflicting. The major candidate gene, *pvmdr1*, was initially posited as a mediator of CQR due to its high sequence homology with the orthologous *pfmdr1*, which is involved in CQR in *P. falciparum*[10]. However, subsequent studies have produced contrasting reports, showing no consistent correlation between *pvmdr1* and CQR[11]. Profiling of polymorphisms within the *pvmdr1* gene led to the association of the Y976F mutation with CQR due to increased chloroquine $IC_{50}$ in samples from Indonesian Papua and Thailand[12]. This mutation has since become a recognised marker for CQR in studies across *P. vivax* endemic regions[9,13,14]. At least 50 *pvmdr1* SNPs have been documented globally, yet none have emerged as a definitive CQR marker, questioning the extent and relevance of *pvmdr1* in modulating CQR. Although profiling the prevalence of *pvmdr1* polymorphisms has increased our understanding of its evolution across different drug pressure backgrounds[15,16], most studies focus solely on the *pvmdr1* gene itself, biasing our understanding of the acquisition of CQR, which is hypothesised to be a multifaceted process involving numerous loci.

Extensive use of antimalarial drugs over several decades has resulted in high-resolution characterisations of recent selection events associated with drug resistance in *Plasmodium* spp. using genome-wide sequence data. Genome-wide analyses have shown evidence for recent positive selection in *P. falciparum* endemic regions with artemisinin resistance[17–19] and revealed selective sweeps around the *pfcrt*, *pfmdr1*, and *pfaat1* genes, which are explicitly linked to CQR[20,21]. Similar selection metrics applied to *P. vivax* genomes have revealed that the orthologous loci associated with antifolate resistance in *P. falciparum* have been subject to selective sweeps[15,16,22]. More recent work has found evidence of selective sweeps around *pvmrp1* in East Asian isolates, a gene associated with chloroquine and mefloquine resistance in the orthologous *P. falciparum pfmrp1*, necessitating further investigation[16].

Understanding the population genetics and dynamics of *P. vivax* malaria, particularly in the context of drug resistance, is essential for successful control and elimination planning. With the expanding repertoire of whole genome sequence data for *P. vivax*, we can now investigate the temporal dynamics of populations pre- and post-chloroquine contraindication, to provide increased insight into the markers of CQR. Here, we leverage publicly available whole genome sequences to present a large-scale population genomics study of *P. vivax*. We provide an expanded insight into population structure, global ancestry, relatedness, and genomic diversity. Using a genome-wide approach, we perform intra- and inter-population analyses between isolates from regions with different degrees of reported CQR to make inferences about both previously described and novel loci that could be mediating CQR and the evolutionary forces that shape *P. vivax* populations.

## Results

### *P. vivax* whole genome sequence data and clonality
A total of 499,206 high-quality bi-allelic SNPs were identified in the non-hypervariable regions of the *P. vivax* genome after filtering,

comprising 1534 isolates from 29 countries. In keeping with previous work[16], we divided these countries into 7 sub-regional populations based on the degree of geographic and genetic proximity: East Africa ($N = 173$), West Africa (1), South America (364), South Asia (156), South East Asia (SEA) (550), Maritime SEA (66), and Oceania (224) (Supplementary Data 1). Similarly, we ascribed each site and each country an overall CQR status adapted from the categories created in a prior meta-analysis of global chloroquine efficacy[5] after pooling all publicly available day 28 recurrence data (Supplementary Data 2 and 3). Here, we describe three categories: chloroquine sensitivity (<5% day 28 recurrences), low-grade CQR (5–10% day 28 recurrences), and high-grade CQR (>10% day 28 recurrences) (Supplementary Data 2 and 3, Fig. S1).

Within-sample diversity, as a metric of multiclonality, was measured using the $F_{WS}$ fixation index, where an $F_{WS} \geq 0.95$ indicates an infection predominated by a single genotype. Here, a large proportion of isolates (71.7%) were monoclonal ($F_{WS} \geq 0.95$). Regionally, mean $F_{WS}$ was greatest in Maritime SEA (0.96), compared with South Asia (0.92), South America (0.92), East Africa (0.91), SEA (0.91), and Oceania (0.86) (Fig. S2). These observations likely reflect trends in transmission intensity, where higher transmission intensity is often marked by higher clone multiplicity.

### Global *P. vivax* isolates form distinct sub-populations to sub-continent level
An analysis of population structure using a SNP-based neighbour-joining (NJ) tree (Fig. 1a) and principal component analysis (PCA) approach (Fig. 1b) applied to 499,206 bi-allelic SNPs revealed *P. vivax* isolates form distinct and independent sub-populations, reflective of their continental or sub-continental origins. In the PCA, the first principal component separated East Asian and Oceanian populations from South American, South Asian, and African populations, producing three major geographical population centres, with South American populations being the most distinct. The number of highly differentiating SNPs ($F_{ST} \geq 0.99$) was positively correlated with geographic distance (Spearman's Coefficient = 0.64) (Supplementary Data 4). These findings suggest gene flow between neighbouring territories is contributing to this phylogeographic distribution and is supported by known waves of human migration[23].

An ADMIXTURE ancestral analysis inferred that global *P. vivax* isolates descend from ten ancestral populations (K1-K10), comprising two in East Africa (K1 and K5), one in South Asia (K1), three in South America (K2, K3, K9), three in SEA (K1, K8, K10), two in Maritime SEA (K4 and K6), and one in Oceania (K7) (Fig. 1c, d). There was some concordance between ADMIXTURE populations and country of origin, where several ancestral populations were made up of predominantly (>95%) one country, including K2 (Panama), K3 (Brazil), and K4 and K6 (Malaysia) (Supplementary Data 1, Fig. S3). Except for K1, which was characterised by African, South Asian, and SEA isolates, all ancestral populations comprised isolates from the same sub-regional grouping. This population structure was supported by a PCA plot coloured by dominant ancestry (Fig. S4).

### Pairwise relatedness supports model of isolation by distance
To further dissect the global *P. vivax* population structure, pairwise relatedness of isolates was inferred by calculating identity-by-descent (IBD), which describes shared evolutionary history. Countries which presented the highest median pairwise IBD include Panama (0.97), Mexico (0.23), and Malaysia (0.19), suggesting reduced outcrossing within these populations (Supplementary Data 5 and 6). While the fractional IBD values of Malaysian and Mexican populations approach the expected value (0.25) of half-siblings in an outbred population, the value for Panama is inflated due to the persistence of a single clone in the region for a decade[24]. Within the remaining subpopulations, IBD sharing was low (<6%), revealing that samples were predominantly

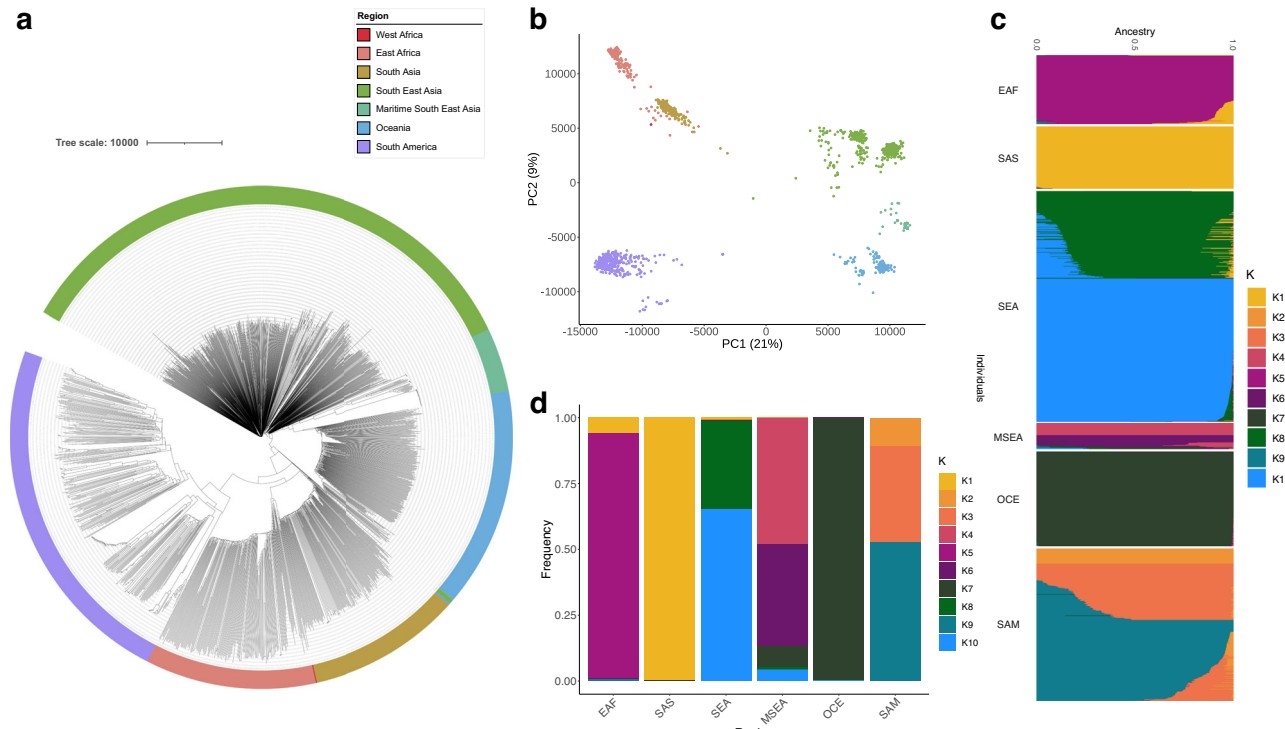

**Fig. 1 | Population structure and ancestry in 1534 global *P. vivax* isolates.** *P. vivax* isolates form distinct populations to sub-continental level. **a** Neighbour-Joining tree for 1534 isolates, constructed using a distance matrix based on 499,206 high-quality bi-allelic SNPs, and coloured based on sub-regional grouping. **b** Principal Component Analysis (PCA) plot of the 1534 isolates, with colours based on sub-regional groupings in (**a**). **c** ADMIXTURE inference of 10 ancestral populations (*K* = 10) in the global dataset, visualised by bar plot, coloured by *K* population grouping, and summarised as frequencies in (**d**). EAF East Africa, MSEA Maritime South-East Asia, OCE Oceania, SAM South America, SAS South Asia, SEA South-East Asia.

weakly related, with a minority of very highly related samples (Fig. S5). Regionally, median pairwise IBD was highest in Maritime SEA (0.15) and lowest in SEA (0.01). In agreement with $F_{ST}$ observations, there was a moderate correlation between inter-regional median pairwise IBD and geographic distance (Spearman's Coefficient = −0.39) (Supplementary Data 4), consistent with the model of isolation by distance[25] which posits that populations in closer geographic proximity have increased genetic similarity.

### IBD sharing reveals putative selective sweeps at *P. vivax* drug resistance loci

To investigate patterns of shared ancestry intrachromosomally, we analysed genome-wide IBD fractions calculated across 10 kb sliding windows, investigating specifically genomic regions falling in the top 1% of fractions (Fig. S6). Due to the high genetic relatedness of isolates from Malaysia, Mexico, and Panama (Fig. 1a, b) and their inflated pairwise IBD values (Fig. S7, Supplementary Data 5), we excluded them from further analysis. Overall, regions with the highest IBD fractions spanned antigenic loci (*pvmsp1*, PVP01_0728900; *pvmsp5*, PVP01_0418400; *pvdbp*, PVP01_0623800), genes involved in life-cycle specific processes (*pvlisp2*, PVP01_0304700), and candidate drug resistance loci (*pvmrp1*, PVP01_0203000; *pvdhfr*, PVP01_0526600; *pvmdr1*, PVP01_1010900; *pvdhps*, PVP01_1429500) (Supplementary Data 7).

The strongest signals (0.35) were observed in Indonesia on chromosome 10 between 320 and 330 kb (Supplementary Data 7), a region encompassing two conserved *Plasmodium* proteins of unknown function (PVP01_1007200 and PVP01_1007250), the former being a pseudogene. This region is of particular interest as it is ~140 kb downstream of *pvmdr1*, the gene putatively responsible for CQR in *P. vivax*. This peak of IBD sharing at 320–330 kb is also found within a large region of high IBD values spanning 200–500 kb on chromosome

10 (Figs. S8 and S9, Supplementary Data 8). Although peaking at 320–330 kb, the median fractional IBD value of the region was 0.12, and the IBD value at *pvmdr1* was 0.11. Interestingly, we found high proportional IBD sharing at this downstream *pvmdr1* locus (320–330 kb; PVP01_1007200, PVP01_1007250) in isolates from PNG (0.19), Sudan (0.13), and Brazil (0.11) (Fig. S10). As with Indonesian isolates, in the Sudanese population, this was found within a region spanning 280–500 kb, with a median IBD value of 0.17. At *pvmdr1* itself, countries with high fractional IBD values were Sudan (0.26), Peru (0.19), Ethiopia (0.19), and Brazil (0.13).

The *pvdhps* and *pvdhfr* genes, located on chromosomes 14 and 5 respectively, are associated with resistance to the antimalarial combination therapy sulfadoxine-pyrimethamine (SP). While SP is not routinely used to treat *P. vivax* malaria, mutations in *pvdhps* and *pvdhfr* structurally align with resistance-conferring mutations observed in their orthologous genes, *pfdhps* and *pfdhfr*, in *P. falciparum*. We observed a trend of differential IBD values surrounding the genes, where isolates had high fractional IBD for one of the SP-resistance genes, but not the other. Isolates from Eritrea, Indonesia, and Myanmar had much greater fractional IBD values at *pvdhps* (0.25, 0.22, and 0.21, respectively) than at *pvdhfr* (0.11, 0.09, <0.01, respectively) (Supplementary Data 7).

Finally, we observed high fractional IBD values in *pvmrp1*, a gene on chromosome 2 associated with resistance to chloroquine, primaquine, and mefloquine in the orthologous *P. falciparum pfmrp1*[26] in Vietnamese (0.22), Thai (0.21), Cambodian (0.18), Burmese (0.16), and Colombian (0.14) isolates. We found no evidence of high proportional IBD sharing in any population at the resistance candidates *pvcrt-o* (PVP01_0109300), *pvpm4* (PVP01_1340900), *pvk13* (PVP01_121100), or in the sister genes to *pvmrp1* and *pvmdr1*, *pvmrp2* (PVP01_1447300) and *pvmdr2* (PVP01_1259100). However, there was a signal from *pvaat1* (PVP01_1120000), the orthologue of a gene newly implicated in CQR in *P. falciparum*[21], in isolates from PNG (0.06).

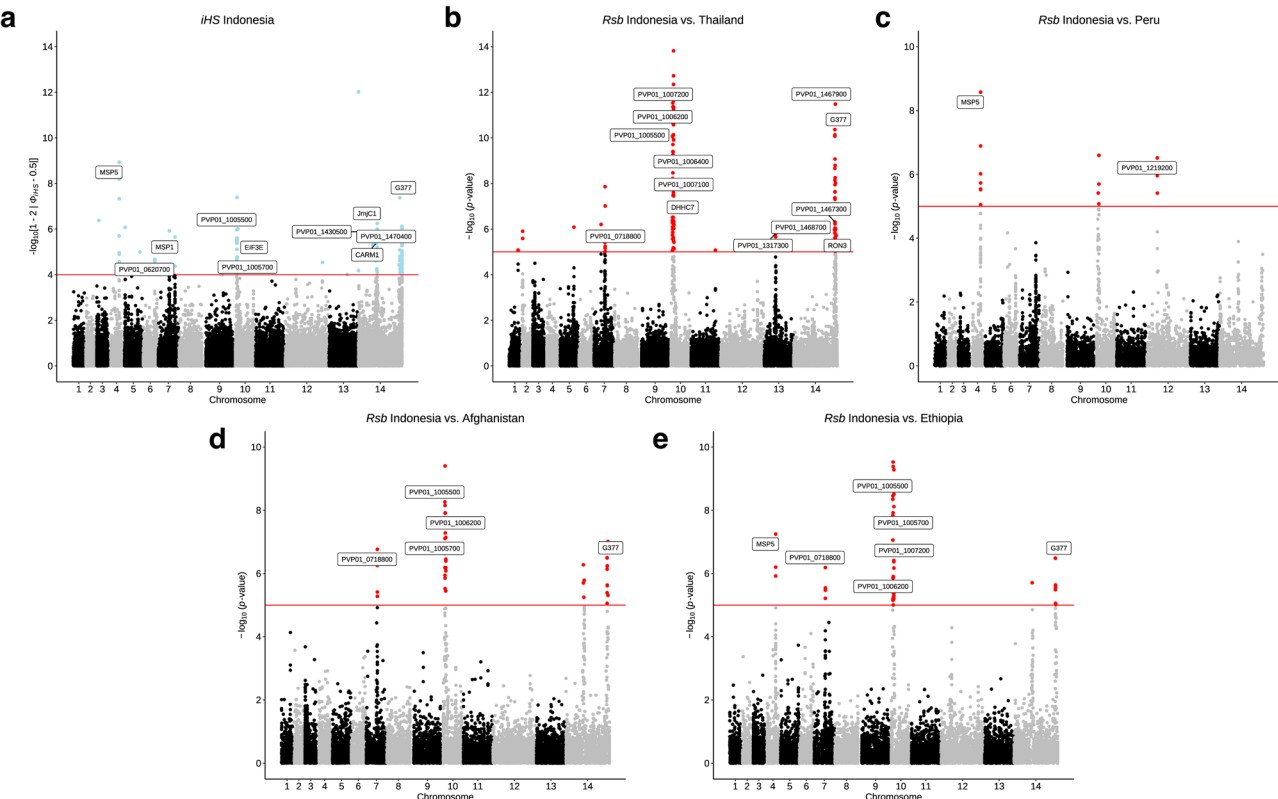

**Fig. 2 | Evidence of selective sweeps on chromosome 10 at downstream *pvmdr1* locus.** Selection at this locus is driving differentiation of Indonesian Papua isolates from *P. vivax* isolates across South-East Asia, South Asia, South America, and East Africa. Manhattan plots showing integrated haplotype homozygosity scores (*iHS*) for SNPs in **a** Indonesian isolates ($N = 106$) and the cross-population test, *Rsb*, showing SNPs under differential selection between Indonesian and **b** Thai ($N = 119$), **c** Peruvian ($N = 82$) **d** Afghan ($N = 50$) and **e** Ethiopian ($N = 102$) isolates. Loci in critical regions, defined here as SNPs with an *iHS* score of $P < 1 \times 10^{-4}$ or *Rsb* score of $P < 1 \times 10^{-5}$ (two-sided tests), are highlighted in blue (*iHS*) and red (*Rsb*).

## Evidence of selective sweeps in candidate drug resistance loci

A genome-wide scan for the top 1% of genes under recent positive selection in monoclonal isolates was performed using the *iHS* metric (Supplementary Data 9). Multiple strong signals ($N = 125$) were found in the genes encoding the surface proteins *pvmsp1* and *pvmsp5* ($P < 1 \times 10^{-24}$), which are under selection pressure due to their interactions with the host immune response. The greatest values were in Afghanistan ($P < 1 \times 10^{-23}$), Thailand ($P < 1 \times 10^{-14}$), Cambodia ($P < 1 \times 10^{-11}$), and Vietnam ($P < 1 \times 10^{-10}$). Similarly, other hotspots of selection pressure were found in the cytoadherence linked asexual protein (*pvclag*, Ethiopia ($P < 1 \times 10^{-12}$), Thailand ($P < 1 \times 10^{-10}$)), the liver-specific protein 2 (*pvlisp2*, Afghanistan ($P < 1 \times 10^{-14}$), Indonesia ($P < 1 \times 10^{-6}$), Cambodia ($P < 1 \times 10^{-6}$)), ApiAP2 transcription factors (PVP01_1418100, Cambodia ($P < 1 \times 10^{-7}$); PVP01_1440600, Cambodia and Vietnam (both $P < 1 \times 10^{-5}$)), and the invasion proteins *pvrbp1a* (Cambodia, Indonesia, Pakistan; $P < 1 \times 10^{-10}$) and *pvrbp2b* (Afghanistan and India; $P < 1 \times 10^{-5}$). In agreement with our IBD findings, a series of 18 SNPs ($P < 1 \times 10^{-8}$) were under selection in Indonesian isolates downstream of *pvmdr1* on chromosome 10 (243480-319423), with peaks at a SNP upstream a translation initiation factor (PVP01_1005600, $P < 1 \times 10^{-8}$) and a SNP in the PVP01_1007200 gene ($P < 1 \times 10^{-7}$). Isolates from Thailand and Pakistan also had SNPs under selection in PVP01_1007200 ($P < 1 \times 10^{-5}$).

To identify signals of differential selection, the cross-population metric, *Rsb*, was used at both a country and regional level, specifically comparing Indonesia with other countries due to known differences in CQR status. Although PNG is also a region of high-grade CQR, due to limited sample size, we excluded it from cross-population selection analysis. The most common SNPs under differential selection encompassed the cluster of SNPs with high *iHS* scores in the Indonesian population downstream of *pvmdr1* ($N = 248$), differentiating Indonesian isolates from those in SEA ($P < 1 \times 10^{-13}$), East Africa (Eritrea and Ethiopia; $P < 1 \times 10^{-7}$), South Asia (Afghanistan, India, Pakistan; $P < 1 \times 10^{-10}$) and South America (Peru; $P < 1 \times 10^{-6}$) (Fig. 2, Supplementary Data 10–12). Beyond SNPs in merozoite surface antigens, we found evidence of selective sweeps via extended haplotype homozygosity at *pvmrp1* in isolates from SEA (Cambodia, China, Vietnam, Thailand) when compared to isolates from South Asia (Afghanistan and India, $P < 1 \times 10^{-6}$). Most of these SNPs were in the promoter region, except for L1361F and 1232I. We also found SNPs within *pvmrp2* under differential selection between isolates from Vietnam and China, Cambodia, and India ($P < 1 \times 10^{-6}$). Similarly, we observed differential signals in *pvdhps* in SEA isolates (Cambodia, Thailand, Vietnam; $P < 1 \times 10^{-11}$), and in *pvdhfr* in South Asian isolates (Afghanistan, India; $P < 1 \times 10^{-7}$). The SNPs S513R and L845F in MDR1 were under differential selection between Thai isolates compared with Vietnamese ($P < 1 \times 10^{-9}$) and Cambodian isolates ($P < 1 \times 10^{-6}$).

## Sub-regional differences in the frequency of putative resistance mutations in *P. vivax* populations

Identifying mutations in putative drug resistance loci can reflect the epidemiology of transmission in both local and global contexts. We therefore evaluated the prevalence of non-synonymous mutations in all isolates in genes with a putative association to antimalarial resistance in *P. vivax*: *pvmdr1* (33), *pvmrp1* (53), *pvdhfr* (17), and *pvdhps* (20) (Table 1) and several additional genes of interest (Supplementary Data 13). Multiple *pvmdr1* mutations have been previously associated with CQR or reduced chloroquine susceptibility[10,13,27], including S513R, G698S, M908L, Y976F, and F1076L, and all, except S513R, are present in the *P. vivax* PvP01 reference. We observed the frequencies of the

**Table 1 | Prevalence of non-synonymous SNPs in genes that have been associated with drug resistance in *P. vivax***

| Gene | Chr[a] | Position[a] | Amino Acid Change | East Africa (N = 173) | South Asia (N = 156) | SEA (N = 550) | Maritime SEA (N = 66) | Oceania (N = 224) | South America (N = 364) |
|---|---|---|---|---|---|---|---|---|---|
| *pvmrp1* | 2 | 154107/8 | A1606V | | | | | | 5.2 |
| *pvmrp1* | 2 | 154168 | H1586Y | | 5.1 | 2.6 | | | 19.8 |
| *pvmrp1* | 2 | 154215/6 | D1570F | | | | | 14.7 | |
| *pvmrp1* | 2 | 154992 | I1478V | 24.3 | 17.9 | 0.2 | | | 9.9 |
| *pvmrp1* | 2 | 154668 | G1419A | 13.4 | 16.7 | | | | 24.7 |
| *pvmrp1* | 2 | 154831 | L1365F | | | | | 17.9 | |
| *pvmrp1* | 2 | 155305 | L1207I | | | **81.5** | | | |
| *pvmrp1* | 2 | 156208 | E906Q | | | | | | |
| *pvmrp1* | 2 | 156563 | E787D | 0.6 | 3.8 | 3.5 | 1.5 | 1.8 | |
| *pvmrp1* | 2 | 158148 | R259I | 35.3 | 27.6 | 0.6 | | | 11.3 |
| *pvmrp1* | 2 | 158223 | T234M | | | **78.4** | | | |
| *pvmrp1* | 2 | 158272 | Y218D | **82.7** | **64.1** | **92.4** | **98.5** | **87.5** | **60.4** |
| *pvmrp1* | 2 | 158545 | V127I | **82.7** | **60.3** | **92.4** | **98.5** | **87.9** | **65.4** |
| *pvdhfr* | 5 | 1077534 | R58K | | | | | 0.5 | 5.5 |
| *pvdhfr* | 5 | 1077535 | R58S | 37.6 | **62.8** | 2.55 | **59.1** | 6.3 | 61.3 |
| *pvdhfr* | 5 | 1077711 | N117T | 6.9 | **50.0** | 30.7 | **90.9** | **69.2** | 17.0 |
| *pvdhfr* | 5 | 1078180 | N273K | 1.7 | 10.3 | 0.2 | | | |
| *pvdhfr* | 5 | 1077878 | I173L | | | | 56.1 | | 8.52 |
| *pvdhfr* | 5 | 1077530 | F57I | | 0.64 | **78.0** | **89.4** | **71.0** | 0.5 |
| *pvdhfr* | 5 | 1077543 | T61M | | | | 31.8 | 41.5 | 27.5 |
| *pvmdr1* | 10 | 479908 | L1076F* | | 1.9 | 19.5 | 1.5 | | **81.6** |
| *pvmdr1* | 10 | 480412 | L908M* | 0.58 | | | | | 8.52 |
| *pvmdr1* | 10 | 480552 | A861E | | 5.8 | 3.6 | | | 0.82 |
| *pvmdr1* | 10 | 480601 | L845F | | 14.7 | 12.0 | 7.6 | 0.4 | |
| *pvmdr1* | 10 | 481042 | S698G | 27.2 | **51.9** | | | | **86.0** |
| *pvmdr1* | 10 | 481595 | S513R | **74.0** | 37.2 | 14.0 | | | |
| *pvmdr1* | 10 | 481636 | D500N | | | | | | 23.6 |
| *pvmdr1* | 10 | 482473 | V221L | | | | | | 21.2 |
| *pvdhps* | 14 | 1270119 | A647V | 24.9 | 4.5 | 0.2 | | | |
| *pvdhps* | 14 | 1270401 | A553G | | 10.3 | 29.8 | **86.3** | 10.7 | |
| *pvdhps* | 14 | 1270911 | G383A* | **71.1** | **83.3** | 9.6 | 4.5 | 16.1 | 39.8 |
| *pvdhps* | 14 | 1270914 | S382C | | | 0.4 | | | 12.9 |
| *pvdhps* | 14 | 1270915 | S382A | | | 10.9 | | | |
| *pvdhps* | 14 | 1271444 | M205I | 71.7 | 0.6 | **96.2** | 6.1 | 1.8 | **58.5** |
| *pvdhps* | 14 | 1271634 | E142G | **64.7** | | | **59.1** | 10.7 | |

Allele frequencies were calculated in isolates if, at a country level, their frequency was ≥10%. Allele frequency is bolded if ≥50%. East Africa (Eritrea, Ethiopia, Sudan); South Asia (Afghanistan, India, Pakistan); South East Asia (SEA; Cambodia, China, Myanmar, Thailand, Vietnam); Maritime SEA (Malaysia); Oceania (Indonesia, Papua New Guinea); South America (Brazil, Colombia, Mexico, Panama, Peru). Asterisked SNPs are those where the reference has the putative resistance allele. SNP prevalence was calculated solely for isolates with homozygous alternate calls.
*Chr* chromosome
[a]Based on the PvPO1_v1 reference.

putative resistance alleles at 70.9% (698S), 97.9% (908L), 100% (976F), and 73.4% (1076L). The F1076L mutation was observed in all isolates from East Africa and Oceania and had lowest frequencies in South American (81.6%) and SEA (19.4%) isolates. Similarly, the G698S mutation was fixed in isolates from Oceania and Maritime SEA and had lowest frequency in South American (14.0%) isolates. The S513R mutation was found exclusively in isolates from SEA (14.0%), South Asia (37.2%), and East Africa (74.0%). There were lower frequency mutations with regional specificity, including D500N (South America, 24.2%), V221L (South America, 21.1%), and L845F (Across Asia and Oceania, 9.5%). Only one isolate from Indonesia had a non-synonymous SNP in *pvmdr1* (L845F).

Mutations in the *pvmrp1* orthologue, *pfmrp1*, decrease susceptibility to chloroquine[26] (H191Y, I876V, T1007M, F1390I), piperaquine[28] (H785N, I876V, T1007M), artemisinins[29] (I876V, F1390I), and antifolates (K1466R). Although there were several *pvmrp1* mutations approaching fixation globally (V127I; 81.2%, Y218N; 80.4%), most mutations showed regional specificity. South American isolates had the greatest number of unique *pvmrp1* mutations including A1606V (5.2%), T1525I (4.4%), and C1018Y (3.3%). Other mutations with regional specificity of varying frequencies included T234M (SEA, 78.4%), L1365F (Oceania, 17.9%), F560I (East Africa (8.1%) and South Asia (1.9%)). We observed a previously undescribed mutation, D1570F, exclusive to Oceanian isolates (Indonesia (16.5%), PNG (10.0%)) (Table 1). Predictions of the domain structure of PvMRP1 revealed two AAA+ ATPase domains, with nucleotide binding domains at residues 647–774 and 1475–1642. Notably, the Oceanian-specific D1570F mutation falls within the latter domain. In silico modelling of PvMRP1, aligned with its orthologous PfMRP1, indicated that several mutations, including PvMRP1 L1365F, are located near known resistance-conferring mutations in PfMRP1, such as the PfMRP1 F1390I mutation associated with artemisinin and chloroquine resistance (Fig. S11). Based on primary structure predictions of PvMRP1, the L1365F mutation is the only Oceania-specific mutation that resides within a transmembrane helix

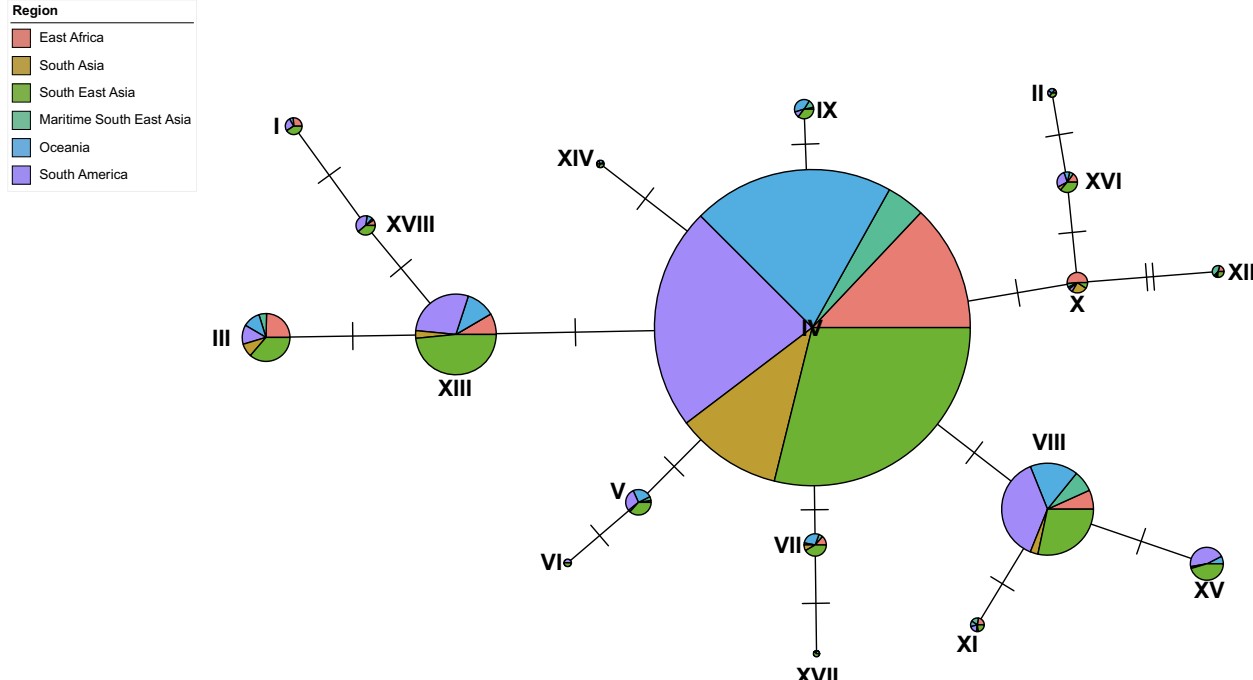

**Fig. 3 | Median joining haplotype network constructed using *pvmdr1* gene sequences from 1238 global isolates.** Each node represents a unique haplotype. Segments within nodes represent isolates from the 6 different subregional groupings and coloured accordingly. Node size is in proportion to the number of samples represented by that haplotype. The number of ticks between nodes represents the number of SNP differences between the two haplotypes.

(TMH), specifically TMH11 (Fig. S12). Outside of Oceanian populations, the only other PvMRP1 mutation found within a TMH is the L1361F mutation, a low-frequency variant present in three South Asian isolates (Afghanistan (1, 1.6%), India (2, 4.2%)) and three SEA isolates (Myanmar (1, 3.6%), Thailand (2, 1.1%)).

Antifolate resistance is attributed to mutations in DHFR and DHPS. In *pvdhfr* and *pvdhps*, numerous SNPs structurally correspond to resistance conferring mutations in *P. falciparum*. The *pvdhfr* S58R and S117T mutations and the *pvdhps* A383G mutation were present in the PvP01 reference, so we report the frequencies of the wild-type (reference) allele (Table 1). The triple *pvdhfr* mutants F57L-S58R-S117N (LRN) coupled with the *pvdhps* double mutant A383G-A553G (GG) are associated with clinical SP failure[30]. We found 3 instances of the *pvdhfr* LRN haplotype in isolates from Myanmar, PNG, and Peru (Supplementary Data 14). The *pvdhps* GG haplotype was globally more prevalent, found in India (33.3%), Indonesia (1.5%), Malaysia (3.2%), and Thailand (1.9%). Another *pvdhfr* haplotype of concern is the *pvdhfr* L/IRMT(F57L/I-S58R-T61M-S117N/T) quadruple mutant, found at moderate prevalence in East Asian and Oceanian isolates: China (16.7%), Malaysia (30.6%), Thailand (65.4%), Myanmar (28.6%), Indonesia (64.9%), and PNG (10.0%). Although we did not observe the *pvdhps* SAKAV mutant (S382A-A383G-K512E-A553G-V585A)[31], we observed the alternate variant, MSAAK (M205I-S382A-A383G-A553G-K512M), exclusive to Thai isolates (4.5%). Finally, although not structurally linked to sulfadoxine resistance, we observed the Q142G mutation in various haplotypes specific to isolates from East Africa (Ethiopia, Eritrea, and Sudan), and the S382C-M205I double mutant specific to isolates from South America (Brazil, Guyana, Panama, and Peru).

## A moderate degree of sequence conservation in global *mdr1* haplotypes

Haplotype networks were constructed to visualise the global diversity of *pvmdr1* and explore haplotypes in regions with high-grade CQR, such as Indonesian Papua, against those with high clinical efficacy of chloroquine (Supplementary Data 2). We observed 169 distinct haplotypes, of which 93 (55.1%) were singletons and 151 (89.3%) had less than 10 observations. A median-joining haplotype network was estimated for the remaining 18 (10.7%) haplotypes identified across 1238 samples (Fig. 3). Haplotype IV, representing the 598P synonymous mutation, was the most prevalent, present in all geographic subregions (41.1%). Most Oceanian isolates (80.8%) were represented by this major haplotype, indicating that isolates from low and high-grade CQR regions have high *pvmdr1* sequence similarity. Globally, 44.0% of isolates had *pvmdr1* haplotypes comprised solely of synonymous mutations, suggesting a substantial degree of conservation at *pvmdr1*. Intra-region estimates of haplotype diversities ($h$) ranged from 0.33 to 0.93, with Oceanian isolates having the lowest genetic diversity ($h = 0.33$), and South Asian isolates with the highest ($h = 0.93$) (Supplementary Data 15).

## Temporal trends in markers under selection in Indonesian isolates

An ex vivo susceptibility study of *P. vivax* isolates from Indonesian Papua revealed the persistence of CQR between 2005–2018, despite dihydroartemisinin-piperaquine replacing chloroquine in 2004[32]. With conflicting evidence of *pvmdr1*'s role in mediating CQR and maintained phenotypic CQR, we sought to investigate temporal trends in genome-wide signals of differential selection within the Indonesian population. We divided all monoclonal isolates into two sub-populations: pre-2014 ($N = 75$) and post-2014 ($N = 29$), with 2014 being 10 years post contra-indication of chloroquine in Indonesia, and a timeframe we perceive adequate for genotypic changes to occur. Comparing *iHS* data revealed that the hotspot of SNPs downstream *pvmdr1* was present solely in the pre-2014 population (Fig. 4a, b). A scan for the top 1% of genes under differential selection (*Rsb*) between the pre- and post-2014 populations revealed the Y218D and V127I *pvmrp1* mutations had high hits ($P < 1 \times 10^{-8}$) (Fig. 4a–c). Contrastingly, pairwise $F_{ST}$ revealed that the D1570F *pvmrp1* mutation was the most highly differentiating between the two populations ($F_{ST} = 0.34$; Median $F_{ST} = 0.027$) (Supplementary Data 16, Fig. S13). This SNP first occurred in 2 isolates from

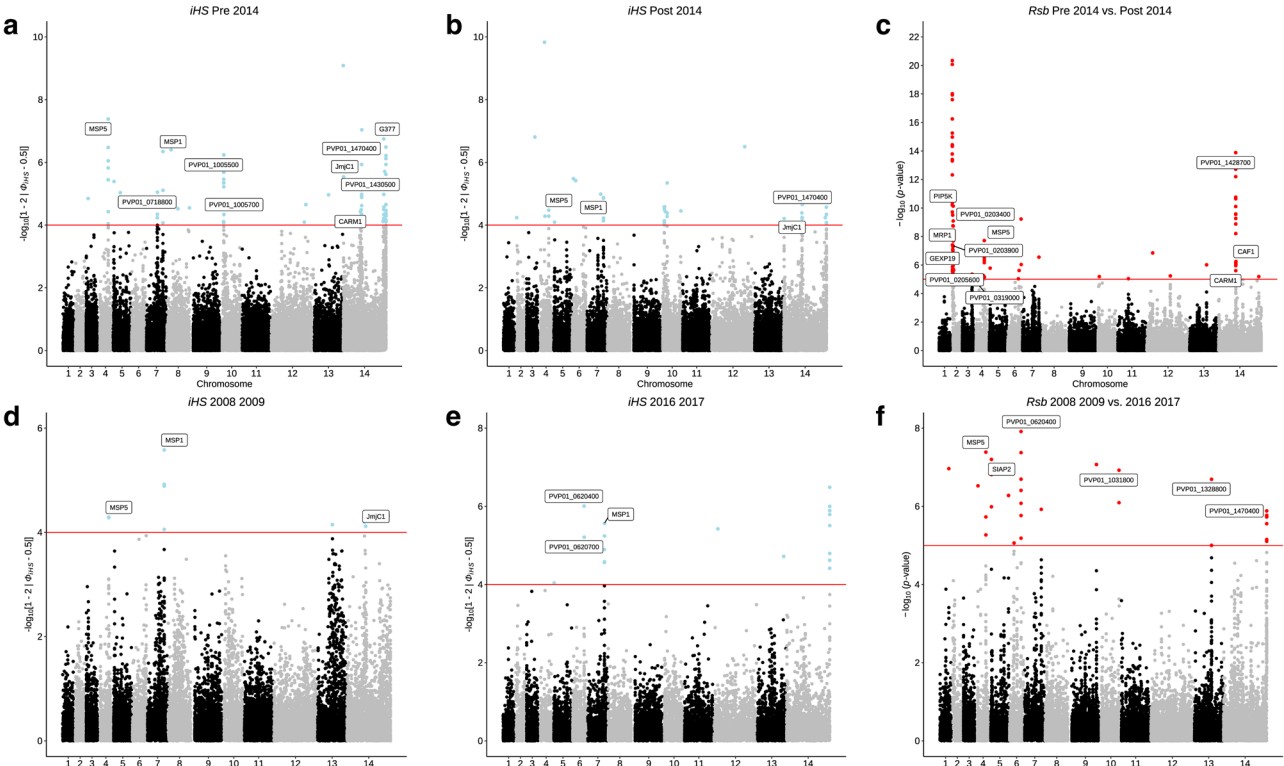

**Fig. 4 | Recent directional selection in Indonesian Papua (N = 104) isolates.** Manhattan plots showing integrated haplotype homozygosity scores (*iHS*) for SNPs in **a** pre-2014 isolates (N = 75), **b** post-2014 isolates (N = 29), **d** 2008–2009 isolates (N = 18), **e** 2016–2017 isolates (N = 10), and the cross-population test, *Rsb*, showing SNPs under differential selection between **c** pre-2014 *vs.* post-2014 isolates and **f** 2008–2009 *vs.* 2016–2017 isolates. Loci in critical regions, defined here as SNPs with an *iHS* score of $P < 1 \times 10^{-4}$ or *Rsb* score of $P < 1 \times 10^{-5}$ (two-sided tests), are highlighted in blue (*iHS*) and red (*Rsb*).

2011, and was present in 58.6% of post-2014 samples, compared with only 20.0% of pre-2014 isolates.

Taking this further, we divided all Indonesian isolates into 5 groups based on year of sample collection (2008–2009 (N = 33), 2010–2011 (N = 49), 2012–2013 (N = 62), 2014–2015 (N = 30), and 2016–2017 (N = 18)) and applied the same intra- and inter-population selection metrics. Across all groups, hotspots of selection were observed on chromosomes 4 and 7, corresponding to *pvmsp5* and *pvsmp1*, and on chromosome 14, corresponding to proteins expressed in gametocytes (G377, PVP01_1467200), proteins for red blood cell adherence (CLAG, PVP01_1401400), and histone methylation machinery (CARM1, PVP01_142800; JmJC1, PVP01_1430400) (Fig. 4d–f, Supplementary Data 17). High *iHS* scores at the downstream *pvmdr1* locus were observed only in populations 2012–2013 and 2014–2015 (all $P < 1 \times 10^{-6}$), indicative of transient directional selection at this locus. Differential selection metrics applied between the divergent 2008–2009 and 2016–2017 groups (4 years *vs.* 12 years post contra-indication of chloroquine) revealed SNPs in the sporozoite-invasion associated protein 2 (*pvsiap2*, $P < 1 \times 10^{-8}$) and the cysteine-rich protective antigen (*pvcyrpa*, $P < 1 \times 10^{-7}$) under differential selection (Supplementary Data 18 and 19). Contrastingly, the $F_{ST}$ metric revealed the *pvmrp1* SNP L1365F was the most highly differentiating ($F_{ST} = 0.54$) between the two populations, with prevalence of the SNP increasing from 0% in 2008–2009 to 38.5% in 2016–2017 (Fig. 5).

## Discussion
Understanding the epidemiological and evolutionary dynamics of malaria using genome-wide data can provide essential biological insights that can be harnessed in the global malaria control and elimination agenda. This is of particular importance in the context of *P. vivax* malaria, where the biological determinants of resistance to the front-line treatment, chloroquine, are still unknown. Here, using

genome-wide SNPs, we were able to show distinct subpopulations of *vivax* isolates to sub-continent level, with classic signs of population differentiation by geographic separation. By profiling SNPs at candidate resistance genes and across the *P. vivax* genome, we have identified new loci exhibiting population differentiation between isolates from Indonesian Papua—a region with a longstanding history of high-grade CQR—and those from chloroquine-sensitive regions. We also describe the temporal dynamics of SNPs near *pvmdr1*, a gene proposed as a determinant of CQR, in Indonesian Papua isolates, and offer hypotheses for the absence of selection pressure within the gene itself over a decade after chloroquine contraindication. Additionally, our selection and temporal SNP dynamics analyses suggest that *pvmrp1* is a promising molecular research and surveillance target. Further investigation is required to fully understand its contribution to antimalarial resistance in *P. vivax*.

Exploration of the population structure of global *P. vivax* isolates revealed the genetic structure largely reflects geographic structure, with significantly higher $F_{ST}$ values between more spatially disparate populations, consistent with previous findings[15,16,33–35]. This observation also agrees with a model of isolation by distance, which indicates increased gene flow and genetic similarity between isolates that are more geographically proximal. At a local scale, we observed that intra-subcontinental populations were less structurally defined, except for South American and, to a lesser extent, SEA populations. These observations are not only a reflection of the complex geographic and ecological niches within these subcontinents, but in a South American context, could be reflective of independent introductions of *P. vivax* in the region, as seen in *P. falciparum*[36]. ADMIXTURE analysis of nuclear SNP data revealed moderate shared South Asian ancestry in East African isolates from Eritrea, Uganda, Sudan, and Madagascar, which has been previously described[16,37], and coincides with human migration of South Asians to East African regions outside the Horn of Africa[38].

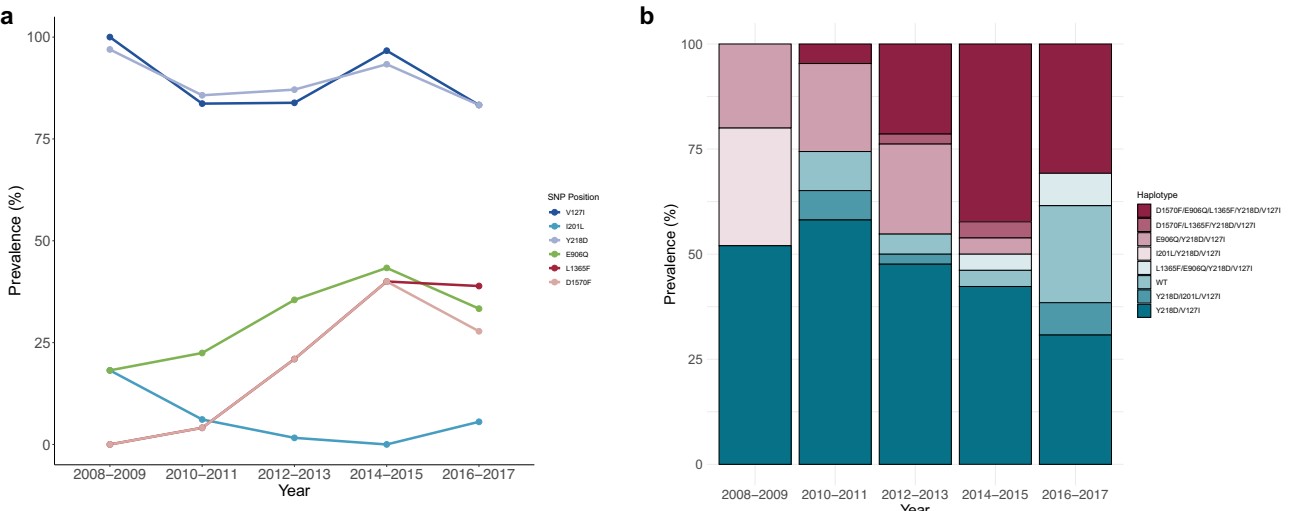

**Fig. 5 | Temporal trends in *pvmrp1* haplotypes in Indonesian Papua (*N* = 104) isolates. a** Change in frequency of non-synonymous mutations in *pvmrp1* across five time periods (2008–2009, 2010–2011, 2012–2013, and 2016–2017). **b** Proportion of major *pvmrp1* haplotypes from the same time periods.

However, this did not reveal an Ethiopian-specific ancestral population, which agrees with prior observations of Ethiopian populations being less structured and more diversified[35]. In the SEA subgrouping, Cambodian and Vietnamese isolates were genetically and ancestrally distinct from isolates from Thailand, Myanmar, and China, which has recently been documented[39]. This is likely a result of the intimate historical links between the two countries throughout the 20th century.

Genome-wide IBD analyses provided helpful insights into the genomic architecture of global *P. vivax* populations, especially in the East African context. We observed, across all three major populations, high fractional IBD at *pvdhps* (Eritrea) and *pvdhfr* (Ethiopia, Sudan). This is most likely a relic of the intensive use of SP as a first-line drug in the treatment of *P. falciparum* malaria, which is co-endemic with *P. vivax* in the region, and due to inadvertent SP drug pressure on *P. vivax* isolates by seasonal malaria chemoprevention regimes[40]. When we paired these data with our SNP profiling results, we observed low frequencies of single and double mutants in *pvdhfr* (S58R, S117N/T) that structurally corresponded with residues linked to pyrimethamine resistance in *P. falciparum*, directly contrasting prior findings of *pvdhfr* double mutants approaching fixation in Ethiopian *vivax* populations[41]. Similarly, in *pvdhps*, we observed an abundance of the wildtype alleles (553A/383G) or other *pvdhps* mutants that have not yet been associated with sulfadoxine resistance (M205I/E142G). The apparent heterogeneity in *pvdhfr/pvdhps* mutants, coupled with clear evidence of *P. falciparum* treatment strategies shaping the *vivax* resistance landscape necessitates further investigation in the East African context, especially due to the moderate rise in *P. vivax* cases in the region between 2019–2022[1].

While in vitro susceptibility studies have shown a role of the Y976F mutation of PvMDR1 in modulating CQR in isolates from Indonesian Papua[12], we found this mutation fixed in global populations (100%). Our observation agrees with no significant change found in chloroquine sensitivity in *P. cynomolgi* introduced with the Y976F mutation[42]. Similarly, we observed the M908L mutation fixed globally (97.9%). With ex vivo measurements associating M908L with reduced susceptibility to chloroquine, dihydroartemisinin, and mefloquine[43], if M908L and Y976F do mediate CQR, this implies that CQR is intrinsic in *vivax* populations. The F1076L mutation, another potential CQR marker, is approaching fixation in *P. vivax* isolates from Sabah, Malaysia[44,45] and Indonesian Papua[10], yet again, as with the Y976F and M908L mutations, we found it at fixation in several countries with high clinical efficacy of chloroquine, such as Afghanistan[46]. The presence of

geographically diverse *pvmdr1* haplotypes with representation of isolates from regions with both low and high clinical efficacy of chloroquine supports the finding of multiple independent local diversification events at *pvmdr1*[47], and strengthens the evidence against these mutations being reliable markers for CQR.

Given the trend of declining chloroquine sensitivity in regions such as Sabah, Malaysia[44], Vietnam[48], and the China-Myanmar border[43], we hypothesised that if *pvmdr1* is mediating CQR, there should be detectable signals of selection in *pvmdr1* indicative of this resistance. We found no evidence of selective sweeps encompassing the *pvmdr1* gene itself within or between any population, despite our genome-wide IBD analyses inferring putative selection events at *pvmdr1* in Sudanese, Ethiopian and Brazilian isolates. Moreover, differential selection of *pvmdr1* SNPs was within the SEA population only. One of the strongest selective sweeps in Indonesian Papua isolates was at a locus proximal to *pvmdr1*, encompassing 18 SNPs in a ~76 kb region. Strong directional selection of this region specific to the Indonesian population at genes PVP01_1007200-250 has been identified previously[49], however, investigations into its role, if any, in mediating CQR have been limited. From IBD data, we could also infer putative selection events at this locus in isolates from PNG and Sudan but limited further investigations due to limited sample size. Here, our study of the temporal dynamics of this cluster of SNPs in the Indonesian context revealed that the selective sweep takes place between 2012 and 2015. Given the temporally transient nature of this genomic signature, several hypotheses can be generated. Firstly, one could hypothesise that these SNPs have no impact on the parasite's fitness or ability to withstand chloroquine pressure. These alleles are therefore neutral and not linked to CQR. Conversely, one could hypothesise that these SNPs reflect changing local dynamics, considering that chloroquine is still readily available in the private sector in Indonesian Papua[50]. In this scenario, the temporal signature could be a result of indirect selection, as the SNPs may be beneficial in periods of increasing chloroquine use, but non-beneficial in periods of increasing dihydroartemisinin-piperaquine use. It is unclear of the role these SNPs play in PNG and Sudan.

Strong signals of positive selection at the *pvmrp1* gene have led to its emergence as a candidate drug-resistance gene[16,51,52]. We found the previously described L1207I SNP under selection in Thai isolates and the *pvmrp1* gene with high fractional IBD in Vietnamese, Thai, and Cambodian isolates. Although not under selection, we observed the mutation D1570F in Indonesian isolates. Investigating the temporal dynamics of this mutation revealed an intriguing pattern. The first

appearance of the corresponding SNP was in an isolate from 2011, and it had strong population differentiation effects between isolates pre- and post-2014. Further investigation of *pvmrp1* between isolates specifically from 2008–2009 to 2016–2017 groupings revealed that along with the D1570F mutation, the L1365F mutation was the most differentiating SNP between the two populations. Dihydroartemisinin-piperaquine has replaced chloroquine as the frontline antimalarial against *P. vivax* in Indonesia, with primaquine still remaining in the treatment strategy to target hypnozoite stages. We observed that, since its introduction, increasing dihydroartemisinin-piperaquine pressure on Indonesian Papuan isolates over time is correlated with a rising prevalence of *pvmrp1* SNPs. Between 2008 and 2018, ex vivo susceptibility of *P. vivax* to piperaquine in the Indonesian Papua population declined[32]. Although clinical efficacy of dihydroartemisinin-piperaquine is reportedly high in Indonesian Papua, therapeutic efficacy studies have shown day 42 recurrence rates ranging from 1.2%[53] to 11.3%[54].

In *P. falciparum*, knockout of the orthologous *pfmrp1* gene resulted in increased susceptibility to piperaquine[26]. Moreover, in a *P. falciparum* genetic cross, progeny with mutant *pfmrp1* were only found in parasites harbouring SNPs in *kelch13* that confer artemisinin resistance[55]. Allele frequency trajectories are not solely driven by selection pressures exerted by drug regimens; changes in the human or vector genomic backgrounds and climate change may also play a role. The appearance of novel *pvmrp1* haplotypes after the introduction of dihydroartemisinin-piperaquine in Indonesia could reflect changing local dynamics, could mitigate the cost of previously fixed alleles, such as those in *pvmdr1*, or could in fact be resistance-conferring. The presence of the PvMRP1 L1365F mutation in TMH11, near the PfMRP1 F1390I mutation associated with chloroquine and dihydroartemisinin resistance, is noteworthy. If PvMRP1 TMHs play an integral role in interacting with antimalarials, mutations at these sites can impact drug influx and efflux, enabling the parasite to evade drug action. Leveraging the *P. cynomolgi* or *P. knowlesi*[56] models to introduce the PvMRP1 D1570F and L1365F mutations and investigating the in vitro efficacy of chloroquine, dihydroartemisinin, and piperaquine will be essential in determining whether drug pressure is the ultimate driver in the appearance of these mutations, and if *pvmrp1* plays a role in mediating resistance.

Overall, the work described here of global *P. vivax* isolates provides insights on population structure, admixture, markers of IBD, differentiation, and selection signatures in the context of drug resistance. Although the markers of CQR remain elusive, our findings of directional selection and selective sweeps in the candidate resistance gene *pvmrp1* highlights the need for broader genotypic and phenotypic surveillance of *P. vivax* to complement elimination efforts.

## Methods

### Sequence data and raw reads processing
Illumina whole genome sequencing data were obtained from the publicly available *Pv4* dataset generated by the MalariaGEN Community Project[57] for *P. vivax* ($N = 1895$) and newly sequenced isolates sent to the UKHSA Malaria Reference Laboratory from imported UK cases ($N = 47$), totalling 1942 isolates available for analysis. After quality control, the final dataset included 1534 (79.0%) isolates from 29 countries: East Africa ($N = 173$, Eritrea (13), Ethiopia (138), Madagascar (4), Sudan (13), Uganda (5)), South America ($N = 364$, Brazil (139), Colombia (58), Guyana (3), Mexico (20), Panama (47), Peru (97)), South Asia ($N = 156$, Afghanistan (63), Bangladesh (1), India (48), Iran (5), Pakistan (37), Sri Lanka (2)), South East Asia (SEA) ($N = 550$, Cambodia (218), China (12), Laos (2), Myanmar (28), North Korea (1), Thailand (179), Vietnam (110)), Maritime SEA ($N = 66$, Malaysia (62), The Philippines (4)), Oceania ($n = 224$, Indonesia (194), PNG (30)), and West Africa ($N = 1$, Mauritania (1)).

### Variant calling and quality control
All raw Illumina sequencing reads were first trimmed using *trimmomatic*[58] (v0.39, parameters PE -phred33, LEADING:3, TRAILING:3, SLIDINGWINDOW:4:20 MINLEN:36). Trimmed reads were aligned to the reference genome, PvP01_v1[59] using *bwa-mem* (v0.7.17, default parameters)[60]. The resultant BAM files were processed with *samtools* functions *fixmate* and *markdup* (v1.9, default parameters)[61]. A training-set of high-quality *P. vivax* single nucleotide polymorphisms (SNPs) from previously published literature was used to calibrate variant calling[16]. Using this set, the GATK BaseRecalibrator and ApplyBQSR functions[62] were run to produce improved BAM files for all isolates[15,16]. SNPs and indels were identified using the GATK HaplotypeCaller function (v.4.4.0.0, parameters: -ERC GVCF) to produce individual sample variant call format (VCF) files. The resultant VCFs were merged to create a multi-sample VCF using the GATK CombineGVCF function (default parameters). A total of 3,671,958 unfiltered SNPs were identified across 1942 isolates. The merged VCF was filtered iteratively (described in full here[16]) to produce the final dataset for subsequent analyses. Briefly, variants identified in the hypervariable subtelomeric regions were removed by mapping the 14 chromosomal sequences against the *P. vivax* PvP01_v1 reference, leaving variants only within the core *P. vivax* genome. Variants with a Variant Quality Score Log-Odds (VQSLOD) score <0, representing variants that are most likely false, were excluded. Variants where >40% of SNPs had missing genotype data were excluded. Monomorphic SNPs, heterozygous SNPs, and indels were also excluded. A set of 175 isolates from Indonesia that passed quality control, were taken from the original Pv4 release VCF (ftp://ngs.sanger.ac.uk/production/malaria/Resource/30) and merged into the VCF created within the present study. The final dataset encompassed 499,206 high-quality bi-allelic SNPs across 1534 samples. SNPs were annotated with predictions of their downstream coding effects using the *SnpEFF* (v5.1) software[63].

### Chloroquine resistance status designation
To ascribe a chloroquine resistance status to each site and each country in this study, we extended the work within a systematic review of Price et al.[5] and used the Worldwide Antimalarial Resistance Network's (WWARN) Vivax Surveyor (http://www.wwarn.org/vivax/surveyor/#0). The Vivax Surveyor houses a repository of all *P. vivax* clinical trial data, including studies from Price et al.'s systematic review and meta-analysis on global chloroquine resistance. We downloaded all *P. vivax* clinical trial data from 1950 to 2019 ($N = 260$) from the WWARN Vivax Surveyor. First, we filtered out studies without a chloroquine arm, resulting in 215 studies. We then excluded studies not conducted in countries with publicly available sequence data in our dataset, reducing the number to 191. Finally, we excluded studies from countries within our dataset with fewer than 10 publicly available *P. vivax* genome sequences or those certified malaria-free by the WHO, leaving a total of 169 studies for our analysis.

To extend the work of Price et al.[5] and include studies published post-2019, we applied an identical research methodology. We systematically searched in the PubMed, Embase, Web of Science, and the Cochrane Database of Systematic Reviews databases, using the same search terms for prospective studies of chloroquine treatment of *vivax* malaria published in English, filtering for "vivax" in the title or abstract. This time, we screened for studies published between January 1, 2019, and June 28, 2024. Estimates of chloroquine efficacy for each site in each study, defined as the proportion of patients with recurrent *P. vivax* parasitaemia at day 28, were extracted, and 95% confidence intervals were calculated using the Wilson score interval procedure. Together with the existing pre-2019 data, a pooled 95% confidence interval was calculated for each site and each country. Based on these pooled day 28 recurrence rates and 95% confidence intervals, we ascribed one of three chloroquine resistance (CQR) statuses to each country, based on and adapted from the a priori categories created by

Price et al.[5]: (i) High-Grade CQR: greater than 10% recurrences by day 28 (with a lower 95% CI of >5%), corresponding to Price et al.'s[5] Category 1; (ii) Low-Grade CQR: between 5 and 10% recurrences by day 28, corresponding to a combination of Price et al.'s[5] Categories 2 and 3; (iii) Chloroquine Sensitive (CQS): less than 5% recurrences by day 28 with chloroquine monotherapy (no primaquine given before day 28) in trials of at least 10 patients, corresponding to Price et al.'s[5] Category 4.

## Population genetics analyses

A binary matrix of pairwise genetic distances was constructed from the filtered biallelic VCF in *PLINK* (v1.9, default parameters)[64]. Using the distance matrix, population structure was assessed by conducting a Principal Component Analysis (PCA) and constructing a neighbour-joining (NJ) tree using the R package *ape* (v5.7). The biallelic VCF of 499,206 SNPs was filtered using *bcftools*[65] (v1.17) *view* function and used to select SNPs with a minor allele frequency (MAF) > 1%, producing a multi-sample VCF of 115,022 high-quality bi-allelic SNPs. The NJ tree was visualised in iTOL[66]. To further investigate population structure, we used the ADMIXTURE software (v1.3.0)[67], a tool used to estimate individual ancestries and population allele frequencies in SNP genotype datasets. *PLINK* (v1.9)[64] was used to convert the MAF-filtered multi-sample VCF to a BED file. Using ADMIXTURE, the most likely number of subpopulations ($K$) was obtained using cross-validation error of 1–10 dimensions of eigenvalue decay. The output was visualised in *R* (v4.2.3). The population differentiation metric, fixation index ($F_{ST}$), was calculated pairwise to assess SNPs driving allele frequency differences between populations at a country and regional level using the *vcftools* (v0.1.17)[68] function -*weir-fst-pop*.

Within-infection genomic diversity or multiplicity of infection, expressed as the variant of inbreeding coefficient $F_{WS}$, was calculated at an individual level using the *R* package *moimix* (v0.0.2.9001, https://github.com/bahlolab/moimix). The $F_{WS}$ metric expresses the probability of the heterozygosity of parasites within an individual against the heterozygosity within a parasite population. Samples with an $F_{WS}$ score of ≥0.95 are highly considered to be monoclonal, whereas samples with an $F_{WS} < 0.95$ suggests mixed strain infections, and therefore, a poorly defined population sub-structure. In all selection analyses, only monoclonal isolates ($F_{WS} \geq 0.95$) were considered.

## IBD and directional selection analyses

Relatedness between all samples was explored in a pairwise manner through identity by descent (IBD) analysis, conducted by the *hmmIBD*[69] package (v2.0.4, default parameters). *hmmIBD* implements a hidden Markov model for inference of pairwise IBD between haploid genotypes, enabling detection of DNA segments with shared ancestry. Only populations of monoclonal samples ($F_{WS} \geq 0.95$) at both a country and regional level with >10 isolates and biallelic SNPs with a MAF > 1% were used for analysis. An additional filtering step to the binary SNP matrix replaced all missing calls to a reference call and all mixed calls to alternative calls. Using sliding windows of 10 kb, IBD was cumulatively calculated and plotted by chromosomal location in *R* (v4.2.3).

To detect signals of recent positive selection, all samples were screened in a pairwise manner at both a country and regional level using the *R* package *rehh* (v.3.2.2)[70] on SNPs with MAF > 1%. We calculated the summary statistics integrated haplotype homozygosity score (*iHS*)[71] for within-population selection, and the cross-population ratio of extended haplotype homozygosity (EHH) expressed as *Rsb* and *XP-EHH* for differential selection between populations[72]. For *iHS* analysis, we describe the ancestral and derived alleles as the reference and non-reference alleles respectively. A positive *iHS* score suggests that the reference allele has undergone selection, whereas a negative *iHS* score suggests selection of a non-reference allele. Critical loci were identified using 10 kb sliding windows, which included at least 5 SNPs with a $p$ value $< 1 \times 10^{-4}$ for *iHS* and $<1 \times 10^{-5}$ for *Rsb* and *XP-EHH*. These cutoffs were calculated using a Gaussian approximation method. *XP-EHH* detects selective sweeps in which a selected allele has approached or achieved fixation in one population while remaining polymorphic in the other population by comparing the lengths of the haplotypes associated with the selected allele in both populations[73]. A positive *XP-EHH* indicates selection occurring in population 1, whereas a negative *XP-EHH* indicates selection occurring in population 2. The *Rsb* metric is the ratio of EHH between two populations, normalised to 1. In-house scripts for analysis are available on GitHub (https://github.com/LSHTMPathogenSeqLab/malaria-hub).

## Haplotype network estimation

The aligned FASTA files for *pvmdr1* and its downstream locus were used to estimate haplotype networks using the pegas (v0.11)[74] package in R, along with nucleotide and haplotype diversity statistics.

## In silico protein structural prediction of PvMRP1 and PfMRP1

MRP1 amino acid sequences from the *P. vivax* P01 (PvMRP1; PVP01_0203000) and *P. falciparum* 3D7 (PfMRP1; PF3D7_0112200) reference genomes were obtained from PlasmoDB[75], and aligned using Clustal Omega[76]. The resultant alignment was visualised in JalView (v2.11.3.3)[77]. The PvMRP1 and PfMRP1 tertiary protein structures were predicted using AlphaFold3 (https://alphafoldserver.com)[78], and aligned and visualised with UCSF ChimeraX (v1.17.3)[79]. The primary structure of PvMRP1 was predicted using DeepTMHMM[80] (https://biolib.com/DTU/DeepTMHMM/), and domain structure predicted using InterPro[81].

## Reporting summary

Further information on research design is available in the Nature Portfolio Reporting Summary linked to this article.

# Data availability

The data used in this study are available at the European Nucleotide Archive (https://www.ebi.ac.uk/ena). For samples from imported *P. vivax* cases diagnosed at the UKHSA Malaria Reference Laboratory, data are under accession codes PRJEB44419 and PRJEB56411. For samples from the MalariaGEN *P. vivax* Genome Variation Project[57], data are under accession codes PRJEB2136, PRJEB2140, PRJEB4409, PRJEB4410, PRJEB10888, PRJNA65119, PRJNA67065, PRJNA67237, PRJNA67239, PRJNA175266, PRJNA240366, PRJNA240531, PRJNA271480, PRJNA284437, PRJNA295233, PRJNA420510, PRJNA432819, PRJNA603279, PRJNA643698, and PRJNA655141. The raw data for 175 Indonesian Papua isolates from the MalariaGEN *P. vivax* Genome Variation Project are available as an unfiltered VCF at ftp://ngs.sanger.ac.uk/production/malaria/Resource/30. All accession codes and sample provenance are detailed in Supplementary Data 1.

# Code availability

For analysis scripts, please see the GitHub repository available at https://github.com/LSHTMPathogenSeqLab/malaria-hub. All scripts use open-source software (see "Methods"). For population genetics analyses, SnpEFF[63] (version 5.1), PLINK[64] (v1.9), ape (v5.7), bcftools[65] (v1.17), R (v.4.2.3), vcftools[68] (v.0.1.17), moimix (v.0.0.2.9001, https://github.com/bahlolab/moimix), ADMIXTURE[67] (v.1.3.0), hmmIBD[69] (v2.04), rehh[70] (v3.2.2), pegas[74] (v0.11). For protein structural analysis we used Clustal Omega (https://www.ebi.ac.uk/jdispatcher/msa/clustalo), AlphaFold3[78] (https://alphafoldserver.com), DeepTMHMM[80] (https://dtu.biolib.com/DeepTMHMM), InterPro[81] (https://www.ebi.ac.uk/interpro/), JalView[77] (v2.11.3.3) and UCSF ChimeraX[79] (v1.17.3).

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

## Acknowledgements

G.C.N-J is funded by a BBSRC LIDo PhD studentship (BB/T008709/1). T.G.C. and S.C. are funded by UKRI MRC (IAA2129, MR/R026297/1, and MR/X005895/1) and EPSRC (EP/Y018842/1), and Wellcome iTPA Translational Accelerator Award (214227/Z/18/Z) grants. C.R.F.M was supported by the São Paulo Research Foundation-FAPESP (2020/06747-4 and 2022/13150-0) and the National Council for Scientific and Technological Development-CNPq (302917/2019-5). J.G.D was supported by fellowships from FAPESP (2019/12068-5 and 2022/02771-3). The SMRU is part of the Mahidol Oxford Research Unit supported by the Wellcome Trust of Great Britain. The funders had no role in study design, data collection and analysis, decision to publish, or preparation of the manuscript.

## Author contributions

T.G.C. and S.C. conceived and designed the study. G.C.N.-J. performed all bioinformatic analyses and interpreted results under the supervision

of T.G.C. and S.C. J.E.P. and E.M. contributed bioinformatic tools and insights. J.G.D., M.S.-M., S.S., G.V.-Z., A.T.C., R.L.D.M., C.R.F.M., D.N., F.N., and C.J.S. contributed samples and sequence data. S.C. sequenced samples. G.C.N.-J. wrote the first draft of the manuscript with inputs from T.G.C. and S.C. All authors provided comments on the versions of the manuscript, approving of the final manuscript.

## Competing interests

The authors declare no competing interests.
