## [Transparent Peer Review file · Nature Communications]

Genomic analysis of global *Plasmodium vivax* populations reveals insights into the evolution of drug resistance

Corresponding Author: Ms Gabrielle Ngwana-Joseph

Version 0:

Reviewer comments:

Reviewer #1

(Remarks to the Author)

The study represents an extension of the authors' previous studies on *P. vivax* population genetics that show positive selection associated with some candidate genes (*pvkelch10*, *pvmrp1*, *pvdhfr* and *pvdhps*, *pvrbp1a*, and *pvrbp1b*) possibly playing some roles in drug responses and parasite invasion (doi: 10.1371/journal.pone.0177134; doi.org/10.1038/s41467-021-23422-3). This study has a larger sample size (1534 vs the previous 558), which potentially can provide more complete and accurate estimates of *P. vivax* population structure, admixture, identity-by-descent (IBD), differentiation, and selection signatures by antimalarial drugs or other factors. The analyses started with PCA and population admixture ancestral analyses to characterize the parasite populations, showing geographic structure (by continent or sub-continent). Then, IBD was calculated, showing high IBD for Panama, Mexico, and Malaysia parasite populations. Next, genome-wide IBD fractions calculated across 10 kb sliding windows were generated to reveal potential loci of selective sweeps. Loci with high IBD included genes encoding antigens, drug resistance candidates, and genes in life-cycle-specific processes. Comparison of genomic haplotypes of isolates from low-grade and high-grade CQR regions revealed potential drug sweeps associated with a locus proximal to *pvmdr1*. Additionally, *pvmrp1* was identified as a candidate for piperazine or primaquine resistance after *Rsb* (extended haplotype homozygosity between populations) analysis of pre-and post-2014 Indonesian Papua isolates. These new data are interesting and important for understanding the molecular basis of selection forces and drug resistance in *P. vivax*. Overall, the analyses in the paper generated many hypotheses for further testing.

The signal from *pvmrp1* is particularly intriguing, although it needs to be clarified what the selection factor is based on the current data. The authors claimed it was a potential marker for piperazine (in the abstract) and primaquine (discussion), but no specific drug data were presented.

The putative SNPs potentially associated with drug responses require further verification. Direct drug assay data from a selected population can be used to provide a more direct association. For instance, two study populations can be set up; one receives CQ treatment, and the other does not. After treatment, test changes in allele frequencies of targeted SNPs. Multiple selection forces could play different roles simultaneously for parasite populations circulating in human populations. Using samples from 'low-grade and high-grade CQR regions' cannot conclusively link a particular SNP to a drug-resistant phenotype. I understand that the authors obtained the genomic data from public databases (pre-existing), and it is not possible at this point to perform drug assays on the samples. I just want to point out this issue. If possible, a smaller-scale study with drug response measurements can be set up to confirm the observation further, as done in a previous study (doi.org/10.1371/journal.pone.0001089). In association studies for human disease genes, separate secondary populations are usually required to confirm the first large-scale association study. Alternatively, candidate mutations can be tested in other *Plasmodium* species, such as *P. cynomolgi* or *P. knowlesi* (again, this is not an easy job either).

Another possibility is that multiple genes, including different transporters, can work together to drive drug resistance to a high level. The authors can consider analyzing multiple genes simultaneously for drug resistance. For example, specific *PvMDR1* alleles may only contribute to CQR when mutations at other genes occur. The authors proposed that a gene (or specific allele) near *Pvmdr1* was associated with CRQ in Papua Indonesian isolates, and *Pvmrp1* was associated with piperazine. Both genes may play a role in responses to both drugs.

Isolates from low-grade and high-grade CQR regions were used to infer CRQ, but no specific drug response levels were provided. There are publications with drug response levels. The authors could perform a meta-analysis using data from related publications. If the results are consistent with the observations here, then the results reported here would be more

convincing.

Some minor points:

- Line 85: The first paper describing the worldwide chloroquine selective sweep on Pfcrt was Wootten et al., Nature 2002, which would be more appropriate to cite here.
- Fig. 1a and 1b are switched in the text.
- Line 359, 'Y976F mutation' should be 'Y976F mutation of PvMDR1'.
- Line 403, 'has' should be 'have.'
- Fig. 1, Fig. 2, Fig. 4, Fig. S1, Fig. S4, and Fig. S5. The labelings are too small.
- Figure legends use capital A, C, and C; the figures use small letters (a, b, c...).
- For the iSH and Rsb plots in Fig. 2 and Fig.4. Did all pre- and post-2014 isolates come from the same locations or matched villages?
- Fig. S8. 'Chromosomal boundaries are denoted by vertical grey dashed lines'. There are no dashed lines in the figure. In the main text, line 170, 'This peak of IBD sharing at 320-330 kb is also found within a large tract of significant IBD values spanning 200-500 kb on chromosome 10'. In the figure, it isn't easy to estimate the positions of the elevated IBD peaks without a vertical line to help identify the positions. The peaks in Fig. S8 appeared to be all under 300k, not 320-330. It may help if the position of pvmdr1 is also marked on the lines.

Reviewer #2

(Remarks to the Author)

Gabrielle et al. performed population structure and selection signal scanning analysis using IBD, iHS, Rsb, and XP-EHH to detect the drug-resistance genes of *P. vivax* with 1,534 parasite whole genome sequences. Similar to what they found before, several parasite candidate genes showed strong selection or association signals for drug resistance. Except for the well-known pvmdr1, another interesting drug-resistance candidate gene, pvmrp1, was highlighted in this genome-wide association study. This study provides meaningful *P. vivax* drug resistance clues to the malaria research. To further verify these candidate genes, it is better to collect more experimental evidence to support the findings or performing a meta-analysis with the published drug response data.

Minor comments/questions:

#1 The use of different software and methods, such as IBD, iHS, Rsb, and XP-EHH, in the study is commendable. However, it would be beneficial to identify common variants that are detected by all or more than two of these methods, as this could provide more robust evidence for your findings.

#2 L471. Why minor allele frequency (MAF) threshold was set as 0.01 rather than commonly used 0.05?

#3 The inclusion of a line connecting each SNP dot to its corresponding gene name in Fig. 2 and 4 would greatly enhance the clarity of these plots. Additionally, a detailed summary table of the top significantly associated SNPs, including information such as SNP chromosome position, variant effect, distance to the candidate genes, allele frequency, and function of the candidate genes, would be a valuable addition to your paper.

#4 For Table 1, the position information is not clear, is it chromosome position? If so, the chromosome name should be added. Mutation type should also be classified, like amino acid change.

#5 Detected top candidate genes can be analyzed further with pathway enrichment analysis. It might reveal some potential drug resistance pathways.

#6 It would be much clearer if a "Locuszoom" figure that enlarges the significantly associated regions, such as chr10:320-330kb, displayed the position of candidate variants or genes in detail was added.

#7 The signals were detected mainly due to the variant allele frequency differences. Drugs can be strong selection pressures, and other factors such as human and vector genomic background and climate changes during these years can also lead to the parasite gene variant allele frequency changes. These points should be included in the discussion part.

Reviewer #3

(Remarks to the Author)

This study makes an important contribution to our understanding of the genomics and evolution of *Plasmodium vivax* malaria. It uses an array of population genetics tools to explore the evidence for recent selection occurring in *P. vivax* populations and focuses on potential cases of selection for drug resistance. The rich dataset and the range of techniques employed in the analysis are strengths of this work. I believe that the importance of the question and the scale of the work make it appropriate for publication in Nature Communications. However, the manuscript would benefit from revisions to improve clarity, and I have some questions about how some of the analysis was done (see below).

1. Readers of the manuscript would benefit from a clear description of historical and current antimalarial drug use in the

regions where the isolates came from, and a summary of what is known about drug resistance in *P. vivax* in these different geographical areas. The manuscript already provides some of this sort of information, e.g. “Indonesian Papua isolates have maintained a CQR phenotype despite the removal of drug pressure exerted by chloroquine since its contraindication in 2004”; “Cambodia and Vietnam...both countries with declining chloroquine sensitivity”. This information is largely concentrated in the Discussion. Having this information summarised in a table (e.g. a table with headings: Country, Number of isolates in this analysis, Antimalarial use history, Evidence for *P. vivax* antimalarial resistance within this country) would be extremely useful for the reader to help them interpret the results (and if some of the information is unavailable that is also useful for the reader to know). The designation of countries as having “low grade” or “high grade” resistance could also be explained in this table, because this definition is unclear in the manuscript.

2. The title includes the phrase “genome wide association study”, which implies a comparison between groups of known properties (e.g. drug resistant and non drug resistant isolates). I suggest not using the phrase “genome wide association study” at all because the actual drug resistance profiles of these specific isolates are not part of this analysis.

3. The abstract states “Differential selection metrics applied between isolates from low-grade and high-grade CQR regions revealed sweeps in a locus proximal to *pvmr1* and in transcriptional regulation genes”. This implies that low grade and high grade CQR regions were specifically compared, but as far as I can tell from the results, all possible comparisons were made between countries and regions (using the *Rsb* metric), and some of the interesting results which emerged showed evidence for differentiation of potential resistance loci between certain “low grade” CQR regions and Indonesia (figure 2). It took me a while to realise this is what the abstract meant so it may be worth re writing this part of the abstract to make it clearer.

4. The Methods do not make clear how statistical analysis of the results were carried out. For example, certain IBD values are referred to as being significant (lines 176, 177, 187). How was this significance determined? Figures 2 and 4 indicate cutoffs for significance – what is the justification for using these particular cutoffs?

5. The Methods refer to in-house scripts for the IBD analysis (line 501). It would be good to indicate to readers how these could be obtained.

5. *pvmr1* is described in the abstract as “as an emerging candidate for piperazine resistance”, but in line 324 as “an emerging putative marker for primaquine resistance”. I am pretty sure, given the rest of the discussion, that line 324 is a typo, however it does raise an interesting point. Evidence from *P. falciparum* (reference [30] in the manuscript) would suggest that mutations in *pvmr1* could be involved in resistance to either piperazine or primaquine. Is the IBD pattern the authors report for *pvmr1* likely to be a consequence of selection from piperazine or primaquine, or both? The likelihood of either drug exerting selection pressure in different geographical regions needs to be discussed more in the manuscript, so that the case for *pvmr1* as a marker of resistance is made more clearly.

The following small points may require correction or alteration to improve clarity

> I found it hard to distinguish the single West African isolate in panels A and B of figure 1. A different colour scheme would be worth considering, so that there is a bigger contrast between the colour for East Africa and the colour for West Africa.

> There are some contradictions between the text and the supplementary tables in terms of whether they refer to the top 5% or the top 1% of the relevant results. E.g. line 156 refers to the top 5% of IBD results but table S5 says top 1%; line 196 refers to the top 1% of genes under recent positive selection measured using the *iHS* metric, but table S7 says top 5%.

> Line 392 of the discussion refers to genes involved in a selective sweep on chromosome 14 in the Indonesian Papua isolates. This part of the discussion suggests the genes include *CLAG*, but *CLAG* is not mentioned in the results relating to this sweep on chromosome 14 (lines 300 and 301).

> Line 359 refers to the Y976F mutation. It would be clearer to insert “in *pvmr1*” to remind readers about the gene in which the mutation occurs.

> The paragraph starting at line 372 was hard for me to follow. Reorganizing the order of the paragraph may make things easier, e.g. starting with the sentence “Considering the trend of declining chloroquine sensitivity [...] we hypothesised that there should be signals of selection in *pvmr1* ...”, then going on to discuss the evidence that was found.

Version 1:

Reviewer comments:

Reviewer #1

(Remarks to the Author)

I accept the responses in the revision and have no further comments.

Line 120-124. Fig.1a and 1b. 1b appears before 1a in the text. They can be switched to have 1a appear first.

Nature portfolio

Competing interests: Both boxes are checked?

Reviewer #2

(Remarks to the Author)

Reviewer #3

(Remarks to the Author)

Thank you to the authors for their revised manuscript. The revisions make their results much clearer, and the new table S2 is a particularly useful addition.

There are two things which I think still need to be clarified before the manuscript is published (points 1 and 2 below), and I also have some additional small comments that I hope will be helpful.

1. The section of the results which describes the IBD analysis is still unclear. I asked about whether the top 1% or top 5% of IBD values were analysed, and the authors have provided tables of both 1% and 5%. However it is still not clear how these are to be used or interpreted by the reader. The text (line 155 onwards) mentions 5% but then refers to table S7 (the top 1% table). It would be clearer to present the results in a way that says something like "genomic regions with consistently high IBD fractions (i.e. >?% in ? countries) included [xyz]". I also wanted to query whether the sentence "we focused on windows with the top 5% of mean IBD fractions, within or around drug resistance loci" is necessary at all? That sentence implies that the authors only considered windows that were already close to drug resistance loci, and yet the rest of the paragraph suggests that they considered all regions, and then found that some regions with the highest IBD values were associated with drug resistance loci (line 162). If my understanding is correct (i.e. they looked at all regions, not just regions already close to drug resistance loci), then lines 155-158 need to be re written for clarity.

2. I asked about the cross population analysis in my earlier review. The authors have now updated the abstract so that it is much clearer- thank you very much for this. Now that the abstract is clear, the section of the text that describes the cross population Rsb analysis (line 212 onwards) is a bit unclear (and in fact, I think the extra sentence added in line 213 makes it less clear!) As I understand it, the authors made all possible pairwise comparisons between countries and *then* interpreted their results in terms of high grade and low grade CQR. The new addition in line 213 implies that the authors already chose the comparisons to make, due to the differences in CQR (but it that was the case, why do they only talk about Indonesia in this section, and not Papua New Guinea which also has high grade CQR?) I think that the logical progression of the ideas in section starting at line 212 should be something like: (i) cross population Rsb analysis was carried out; (ii) Indonesia looked particularly different from other countries in this analysis, then followed by the discussion/potential explanation point that Indonesia happens to have especially high grade CQR. I may have misunderstood, so please re write this section to reflect what the authors want to say, but I wanted to explain why I am still struggling with the way this section has been written.

Small suggestions:

3. The new table S2 is excellent and could be referred to in more places, e.g. line 60 of the current manuscript (currently it looks as though that paragraph needs more referencing - directing the reader to table S2 will help). I could not find China in the table, even though I believe there were >10 samples from China (Table S1). This may simply be an omission and China can be added, but if there is a reason China is not included, this should be explained.

4. The sentence "We therefore evaluated the prevalence of resistance haplotypes (non synonymous mutations) in all isolates..." (lines 230-231) is a bit confusing because the phrase "resistance haplotypes" implies a haplotype that has definitely been associated with resistance, yet the rest of the section refers to non synonymous mutations, some of which have some evidence for an association with resistance and some of which do not (or do not yet!) If I have understood correctly, then it would be clearer to say something like "We therefore evaluated the prevalence of non-synonymous mutations affecting known or candidate antimalarial resistance loci". The subheading (line 228) could also be slightly altered to say "putative" drug resistance mutations (or similar).

5. In line 368, I believe the word "sites" or possibly "residues" is missing, i.e. "structurally corresponded with sites/residues linked to... "

6. Line 431 refers to "continued selection at the upstream pvmdr1 locus" in Indonesian Papua. What is the evidence for continued selection? I was also not clear what the "upstream" region was, since I thought that the region with the potential selective sweep was downstream of pvmdr1.

7. In lines 403 to 404 the authors hypothesise that the region where they have identified a putative selective sweep may contain "neutral alleles". I was not sure what was being argued here. I agree that if there was a temporary selective pressure between 2012 and 2015 it may not have been chloroquine, and I also agree that it's possible that "changing local dynamics" are responsible for the pattern, but I think it needs to be made clearer if the authors are saying that, in fact, the pattern they have found may not indicate any form of selection (i.e. all the variation in the region is neutral), or that they think it still does

indicate some form of selective sweep, but not necessarily one caused by antimalarial drugs.

REVIEWER COMMENTS RESPONSE

REVIEWER 1

Reviewer #1 (Remarks to the Author):

The study represents an extension of the authors' previous studies on *P. vivax* population genetics that show positive selection associated with some candidate genes (*pvkelch10*, *pvmrp1*, *pvdhfr* and *pvdhps*, *pvrpb1a*, and *pvrpb1b*) possibly playing some roles in drug responses and parasite invasion (doi: 10.1371/journal.pone.0177134; doi.org/10.1038/s41467-021-23422-3). This study has a larger sample size (1534 vs the previous 558), which potentially can provide more complete and accurate estimates of *P. vivax* population structure, admixture, identity-by-descent (IBD), differentiation, and selection signatures by antimalarial drugs or other factors. The analyses started with PCA and population admixture ancestral analyses to characterize the parasite populations, showing geographic structure (by continent or sub-continent). Then, IBD was calculated, showing high IBD for Panama, Mexico, and Malaysia parasite populations. Next, genome-wide IBD fractions calculated across 10 kb sliding windows were generated to reveal potential loci of selective sweeps. Loci with high IBD included genes encoding antigens, drug resistance candidates, and genes in life-cycle-specific processes. Comparison of genomic haplotypes of isolates from low-grade and high-grade CQR regions revealed potential drug sweeps associated with a locus proximal to *pvmdr1*. Additionally, *pvmrp1* was identified as a candidate for piperazine or primaquine resistance after Rsb (extended haplotype homozygosity between populations) analysis of pre- and post-2014 Indonesian Papua isolates. These new data are interesting and important for understanding the molecular basis of selection forces and drug resistance in *P. vivax*. Overall, the analyses in the paper generated many hypotheses for further testing.

Thank you very much for your comments, and for highlighting our insights on the population structure, ancestry, and relatedness of this expanded *P. vivax* dataset.

The signal from *pvmrp1* is particularly intriguing, although it needs to be clarified what the selection factor is based on the current data. The authors claimed it was a potential marker for piperazine (in the abstract) and primaquine (discussion), but no specific drug data were presented.

Thank you very much for your insights into the *pvmrp1* signal, and detecting a typo on line 324, which has been corrected along with an updated statement on our conclusions from the *pvmrp1* signals on lines 338-341: "Additionally, our selection and temporal SNP dynamic analyses suggest that *pvmrp1* is a promising molecular research and surveillance target, potentially linked to piperazine resistance. Further investigation is required to fully understand its contribution to antimalarial resistance in *P. vivax*."

Additionally, in our Introduction (L51-60), we are more specific about the drug regimens used in countries with chloroquine resistant *P. vivax* (Indonesia, Papua New Guinea, and Cambodia): "Chloroquine, in combination with primaquine, is the front-line treatment for the radical cure of *P. vivax* malaria in most endemic countries. First documented in the late 1980s, chloroquine resistant (CQR) *P. vivax*, or chloroquine treatment failure, is described by the World Health Organization (WHO) as the presence of parasitaemia on day 28 following treatment. CQR has consequently led to the adoption of artemisinin-based therapies to replace chloroquine in several countries, including dihydroartemisinin-piperazine in Indonesia, artesunate-mefloquine in Cambodia, and artemether-lumefantrine in Papua New Guinea (PNG)^{3,4}. Indonesian Papua and PNG have been the epicentre of high-grade, or category 1 CQR, defined as >10% recurrences by day 28 of treatment⁵. Over the past two decades, increasing reports of CQR *P. vivax* beyond these countries have been characterised by parasite persistence 28 days post-treatment, spurring research into the molecular determinants of CQR." We address the above point on

drug data in your final comment to us and agree that we need to address this explicitly in our manuscript.

The putative SNPs potentially associated with drug responses require further verification. Direct drug assay data from a selected population can be used to provide a more direct association. For instance, two study populations can be set up; one receives CQ treatment, and the other does not. After treatment, test changes in allele frequencies of targeted SNPs. Multiple selection forces could play different roles simultaneously for parasite populations circulating in human populations. Using samples from 'low-grade and high-grade CQR regions' cannot conclusively link a particular SNP to a drug-resistant phenotype. I understand that the authors obtained the genomic data from public databases (pre-existing), and it is not possible at this point to perform drug assays on the samples. I just want to point out this issue. If possible, a smaller-scale study with drug response measurements can be set up to confirm the observation further, as done in a previous study (doi.org/10.1371/journal.pone.0001089). In association studies for human disease genes, separate secondary populations are usually required to confirm the first large-scale association study. Alternatively, candidate mutations can be tested in other Plasmodium species, such as *P. cynomolgi* or *P. knowlesi* (again, this is not an easy job either).

Thank you for this insightful comment. We agree that we cannot conclusively link SNPs to drug-resistance phenotypes without direct drug assay data. We initially highlighted in our Discussion that candidate mutations should be tested in the *P. cynomolgi* or *P. knowlesi* models (L438-442), as these experiments go beyond the scope of our genomics study. Moreover, we agree that testing candidate mutations in *in vitro* models has proven difficult in the past, particularly with *pvmrp1* and *P. knowlesi* (<https://doi.org/10.1371/journal.pntd.0007470>). However, we have now modified our language in both the Abstract and Discussion about these novel *pvmrp1* SNPs in the Indonesian Papua population, firstly removing language that suggests that *pvmrp1* is a definitive marker for CQR, and emphasising an additional need for *ex vivo* susceptibility studies and drug assays to validate these here:

- **Abstract, L32-36: “Selective sweeps in a locus proximal to *pvmrp1*, a putative marker for CQR, along with transcriptional regulation genes, distinguish isolates from Indonesia from those in regions where chloroquine remains highly effective. In 106 isolates from Indonesian Papua, the epicentre of CQR, we observed an increasing prevalence of novel SNPs in the candidate resistance gene *pvmrp1* since the introduction of dihydroartemisinin-piperaquine.”**
- **Discussion, L338-341: “Additionally, our selection and temporal SNP dynamic analyses suggest that *pvmrp1* is a promising molecular research and surveillance target, potentially linked to piperaquine resistance. Further investigation is required to fully understand its contribution to antimalarial resistance in *P. vivax*.”**

Although we cannot conclusively link these mutations to drug-resistance phenotypes, we decided to extend our analysis by performing *in silico* protein structural analysis to see if there is a correlation between known resistance-conferring mutations in the orthologous *P. falciparum* PfMRP1 and the novel PvMRP1 mutations we describe in *P. vivax*. We have added our findings in Results, Methods, and have 2 supplementary figures (S11, S12) showing the aligned PvMRP1 and PfMRP1 protein structure predictions:

- **Results, L253-263: “Predictions of the domain structure of PvMRP1 revealed two AAA+ ATPase domains, with nucleotide binding domains at residues 647-774 and 1475-1642. Notably, the Oceanian-specific D1570F mutation falls within the latter domain. *In silico* modelling of PvMRP1, aligned with its orthologous PfMRP1, indicated that several mutations, including PvMRP1 L1365F, are located near known resistance-conferring mutations in PfMRP1, such as the PfMRP1 F1390I mutation associated with chloroquine and artemisinin resistance (Fig. S11). Based on primary structure predictions of PvMRP1,**

the L1365F mutation is the only Oceania-specific mutation that resides within a transmembrane helix (TMH), specifically TMH11 (Fig. S12). Outside of Oceanian populations, the only other PvMRP1 mutation found within a TMH is the L1361F mutation, a low-frequency variant present in three South Asian isolates (Afghanistan (1, 1.6%), India (2, 4.2%)) and three SEA isolates (Myanmar (1, 3.6%), Thailand (2, 1.1%)).”

- **Methods, L572-L578:** “*In silico* protein structural prediction of PvMRP1 and PfMRP1.
- **MRP1 amino acid sequences from the *P. vivax* P01 (PvMRP1; PVP01_0203000) and *P. falciparum* 3D7 (PfMRP1; PF3D7_0112200) reference genomes were obtained from PlasmoDB⁷⁰, and aligned using Clustal Omega⁷¹. The resultant alignment was visualised in JalView⁷². The PvMRP1 and PfMRP1 tertiary protein structures were predicted using AlphaFold 3⁷³, and aligned and visualised with UCSF ChimeraX⁷⁴. The primary structure of PvMRP1 was predicted using DeepTMHMM⁷⁵ (<https://biolib.com/DTU/DeepTMHMM/>), and domain structure predicted using InterPro⁷⁶.**

These additional *in silico* observations make a clearer case for the involvement of *pvmp1* in antimalarial drug resistance in *P. vivax*, and we predict these SNP effects to be strong, especially in combination with our IBD, *iHS*, *Rsb*, and *XP-EHH* observations.

Another possibility is that multiple genes, including different transporters, can work together to drive drug resistance to a high level. The authors can consider analyzing multiple genes simultaneously for drug resistance. For example, specific PvMDR1 alleles may only contribute to CQR when mutations at other genes occur. The authors proposed that a gene (or specific allele) near *Pvmdr1* was associated with CRQ in Papua Indonesian isolates, and *Pvmp1* was associated with piperazine. Both genes may play a role in responses to both drugs.

Thank you for your insightful comment. Sorry for the confusion, we do not propose that the downstream *pvmdr1* locus was associated with CQR, nor do we believe that to be the case in the evidence we presented. We made the following statements to highlight that they are of interest due to their proximity to *pvmdr1* in the:

- **Results, L168-L169:** “This region is of particular interest as it is ~140 kb downstream of *pvmdr1*, the gene putatively responsible for CQR in *P. vivax*.”
- **Results, L206-L207:** “In agreement with our IBD findings, a series of 18 SNPs ($P < 1 \times 10^{-8}$) were under selection in Indonesian isolates downstream of *pvmdr1* on chromosome 10”.
- **Discussion, L396-L397:** “One of the strongest selective sweeps in Indonesian Papua isolates was at a locus proximal to *pvmdr1*, encompassing 18 SNPs in a ~76kb region.”
- **Discussion, L403-L406:** “Given the temporally transient nature of this genomic signature, we hypothesise that they are either neutral alleles, and therefore not linked to CQR, or instead, reflect changing local dynamics and drug pressures within the country, considering that chloroquine is still readily available in the private sector in Indonesian Papua⁵⁰.”

We use the term “putative” above having set up the context for this, particularly in the Introduction (L62-L64): “The major candidate gene, *pvmdr1*, was initially posited as a mediator of CQR due to its high sequence homology with the orthologous *pfmdr1*, which is involved in CQR in *P. falciparum*⁶. However, subsequent studies have produced contrasting reports, showing no consistent correlation between *pvmdr1* and CQR⁷.” However, to remove any ambiguity, we have modified our Discussion (L335-337) to be more explicit that it is specifically *pvmdr1*, and not the proximal SNPs, that has been recognised through prior studies as a determinant of CQR here: “We also describe the temporal dynamics of SNPs near *pvmdr1*, a gene proposed as a determinant of CQR, in Indonesian Papua isolates”

Lastly, we agree that CQR is most likely being mediated by a combination of mutations in multiple genes. This has been seen extensively in *P. falciparum*, for example, CQR being mediated by mutations in *pfCRT*, *pfPR*, and *pfmdr1*. We have added a new comment on this in our Discussion, on L429-L433: “However, the increasing frequency of

pvmrp1 SNPs after introduction of dihydroartemisinin-piperaquine in the Indonesian population, along with continued selection at the upstream *pvm-dr1* locus, supports the hypothesis that parasite adaptation to drug regimen changes is polygenic in nature, as seen in *P. falciparum* with CQR^{17,18}.” Theoretically, *pvmrp1* and *pvm-dr1* could indeed be playing a role together. However, in our Indonesian Papua dataset, there was only one isolate with a non-synonymous SNP in *pvm-dr1*, which we explicitly stated on line 243. Having only one data point means no meaningful conclusions could be made from that analysis. However, we agree that it would be an interesting avenue of research, and state this in the Discussion on L433-L435: “The appearance of novel *pvmrp1* haplotypes under the changed drug regime could be to mitigate the cost of previously fixed alleles, such as those in PvMDR1, or could in fact be resistance-conferring.”

Isolates from low-grade and high-grade CQR regions were used to infer CRQ, but no specific drug response levels were provided. There are publications with drug response levels. The authors could perform a meta-analysis using data from related publications. If the results are consistent with the observations here, then the results reported here would be more convincing.

Thank you for the insightful comment. We recognise that not including specific drug response levels or incorporating information on day 28 recurrence rates detracts from the results we presented here and agree that incorporating these would make our results more convincing. A systematic review and meta-analysis on the extent of chloroquine resistance (CQR) globally has been published in The Lancet by Ric Price and colleagues (DOI: [https://doi.org/10.1016/S1473-3099\(14\)70855-2](https://doi.org/10.1016/S1473-3099(14)70855-2)). As a result, an online mapping database for *P. vivax* clinical trials was created: The Vivax Surveyor (doi: [10.1016/j.ijpddr.2017.03.003](https://doi.org/10.1016/j.ijpddr.2017.03.003)). The Vivax Surveyor has not been updated since 2019. Price et al.’s study created 4 categories to describe the status of CQR, ranging from Category 1 (>10% Day 28 recurrences), which would be described as high-grade CQR, going down to Category 4 (<5% Day 28 recurrences), which is described as chloroquine sensitivity (CQS). We applied the methodology from Price and colleagues’ 2014 work to all studies from 1st January 2019 to date. This revealed 13 novel studies of the clinical efficacy of chloroquine (Brazil – 1, Ethiopia – 5, Myanmar – 3, Pakistan – 1, India – 1, Vietnam 1). Moreover, when we investigated the sites in which these studies took place within these countries, and cross-referenced with the sites in our analysis dataset, this left 4 novel (post-2019) relevant studies across 2 countries (Brazil – 1, Myanmar – 3). When we compared the day 28 recurrence rates from these post-2019 studies to the pre-2019 studies (i.e., those in the Price et al., 2014 meta-analysis), we found that chloroquine efficacies in these studies at these sites, based on the CQR category they fell in to, matched perfectly. This shows that the findings of Price et al.’s meta-analysis, and the CQR status designations are not only still relevant but remain the same today.

A 2018 genomic analysis of Malaysian *P. vivax* populations by Auburn et al. published in *Nature Communications* (<https://doi.org/10.1038/s41467-018-04965-4>) defined, in their Introduction, low-grade CQR as “5-15% of patients treated with chloroquine having recurrent parasitaemia by 28”, and high-grade CQR as “...more than 50% of patients treated with CQ having recurrent parasitaemia by 28”, essentially showing that there is heterogeneity in what constitutes high-grade vs. low-grade CQR.

For the purposes of our work here, we extended the work of Price et al.’s 2014 study, and defined 3 CQR categories herein, instead of 4, where chloroquine sensitivity was defined as <5% day 28 recurrences, low-grade CQR was 5-10% day 28 recurrences, and >10% was high-grade CQR. This is now stated explicitly in the text (L103-108). However, we wanted to firmly address the great point you raised here, not only to increase the robustness of our work but to help the readers contextualise our findings better. Therefore, we have added 2 additional supplementary tables and a supplementary information file:

- (i) Supplementary Table S2 details:

- **Current and prior antimalarial use in *P. vivax* endemic countries within our analysis dataset**
 - **Any Day 28 recurrence rates for specific sites within these countries**
 - **CQR status designation by Price et al.,**
 - **Pooled day 28 recurrence rates with 95% confidence intervals (CIs).**
 - **While we understand the heterogeneity of CQR at both a site and country-level, we also decided to provide a pooled day 28 recurrence rate at the country-level, based on all pass studies for that given country, and then provide a designation of either: “CQS, Low-Grade CQR, High-Grade CQR, or Unknown”.**
 - **We decided to provide a pooled country-level designation (along with site-level designations) because most antimalarial policy changes are at a country-level.**
 - **The designations we previously gave in our first manuscript submission and the designations in our resubmission are the same, however, we have just added context into how these designations were reached.**
- (ii) **Supplementary Table S3 includes all studies included in the Price et al. 2014 study, and the new studies we found when applying the same criteria to post-2019 studies.**
- (iii) **Supplementary Figure S1 details the methodology/inclusion criteria for studies in which we obtained day 28 recurrence rates (as per Price et al.).**

Some minor points:

--Line 85: The first paper describing the worldwide chloroquine selective sweep on Pfcrt was Wootten et al., Nature 2002, which would be more appropriate to cite here. **Now cited.**

--Fig. 1a and 1b are switched in the text. **Corrected on lines 120, 124, and 159.**

--Line 359, 'Y976F mutation' should be 'Y976F mutation of PvMDR1'. **Corrected on line 376.**

--Line 403, 'has' should be 'have.' **Corrected (line 408).**

--Fig. 1, Fig. 2, Fig. 4, Fig. S1, Fig. S4, and Fig. S5. The labelings are too small. **Corrected**

--Figure legends use capital A, C, and C; the figures use small letters (a, b, c...). **Corrected, now all consistently lowercase.**

--For the iSH and Rsb plots in Fig. 2 and Fig.4. Did all pre- and post-2014 isolates come from the same locations or matched villages?

The published metadata was limited, but isolates were collected from the same set of villages between surveys. In published therapeutic efficacy studies of CQR, across 10 TES/clinical trials at 8 different Papuan sites, all have Category 1 (high-grade CQR) except for in 1 study where primaquine was administered in high-dose at start of the study (high-dose), so in theory could be inferred as CQR (Table S3).

--Fig. S8. 'Chromosomal boundaries are denoted by vertical grey dashed lines'. There are no dashed lines in the figure. In the main text, line 170, 'This peak of IBD sharing at 320-330 kb is also found within a large tract of significant IBD values spanning 200-500 kb on chromosome 10'. In the figure, it isn't easy to estimate the positions of the elevated IBD peaks without a vertical line to help identify the positions. The peaks in Fig. S8 appeared to be all under 300k, not 320-330. It may help if the position of *pvmdr1* is also marked on the lines. **Thanks for spotting that it was the incorrect plot. We have now corrected this figure, which is now Fig. S9, highlighting explicitly the region around 320-330kb +/- 10kb, and have also positioned *pvmdr1*. We have modified the figure legend so that it states: "Boundaries of regions of interest are denoted by vertical grey dashed lines". Moreover, we have added an additional supplemental figure, Fig. S10, showing the IBD fractions across chromosome 10 in its entirety, and then the zoomed in 240-340kb locus. Once again, thank you.**

REVIEWER 2

Reviewer #2 (Remarks to the Author):

Gabrielle et al. performed population structure and selection signal scanning analysis using IBD, iHS, Rsb, and XP-EHH to detect the drug-resistance genes of *P. vivax* with 1,534 parasite whole genome sequences. Similar to what they found before, several parasite candidate genes showed strong selection or association signals for drug resistance. Except for the well-known *pvmdr1*, another interesting drug-resistance candidate gene, *pvmrp1*, was highlighted in this genome-wide association study. This study provides meaningful *P. vivax* drug resistance clues to the malaria research. To further verify these candidate genes, it is better to collect more experimental evidence to support the findings or performing a meta-analysis with the published drug response data.

Thank you for the comments. We agree that further verification of these candidate genes is necessary and have addressed your comments in our response to Reviewer #1, as they also shared this opinion. Briefly, we provided evidence to Reviewer #1 explaining why the results from the last global meta-analysis of chloroquine efficacy are still relevant today, based on us extending their study and updated it with new papers since its publication in 2014 and the corresponding updates to The Vivax Surveyor which accompanies the meta-analysis, and was last updated in 2019. We have created two additional supplementary tables (S2, S3) and added an additional supplementary figure, Fig. S1 which we describe above in our response to Reviewer #1 detailing the drug response data and how we used that to categorise chloroquine resistance status in our dataset.

Minor comments/questions:

1. The use of different software and methods, such as IBD, iHS, Rsb, and XP-EHH, in the study is commendable. However, it would be beneficial to identify common variants that are detected by all or more than two of these methods, as this could provide more robust evidence for your findings.

We agree, and have incorporated a table which shows regions that fall into the top 1% *iHS*, *Rsb*, *XP-EHH*, and IBD scores when making comparisons between Indonesian Papua isolates and isolates from other populations (Table S13).

2. L471. Why minor allele frequency (MAF) threshold was set as 0.01 rather than commonly used 0.05?

We use a MAF threshold of 1% as opposed to 5% because it has been applied in other *P. vivax* publications (<https://doi.org/10.1038/s41467-018-04965-4>, <https://doi.org/10.1038/s41467-021-23422-3>). We are also using a much larger sample population (N = 1,534) than these papers which also use the same MAF threshold, so we believe that 1% is justified.

3. The inclusion of a line connecting each SNP dot to its corresponding gene name in Fig. 2 and 4 would greatly enhance the clarity of these plots. Additionally, a detailed summary table of the top significantly associated SNPs, including information such as SNP chromosome position, variant effect, distance to the candidate genes, allele frequency, and function of the candidate genes, would be a valuable addition to your paper.

While we understand that a line connecting each SNP dot to its corresponding gene name could add clarity to the plot, in our experience, it can clutter the plot too. We have provided Supplementary tables S10, S11, S18, and S19 that are detailed summary tables of all these top significantly associated SNPs, their chromosomal positions, the corresponding variant effect, amino acid changes (if non-synonymous), and the *iHS*, *Rsb* and *logp* values, which addresses your latter point. However, to further supplement this, we have added the products of the genes that these SNPs are found, to help contextualise their role and potential importance for the reader.

4. For Table 1, the position information is not clear, is it chromosome position? If so, the

chromosome name should be added. Mutation type should also be classified, like amino acid change.

Thank you. We agree that this was not clear, and so have specified that these positions pertain to chromosomal position and have changed the table header of “Mutation” to “Amino Acid Change”, along with adjusting the title of the table to be clear that these mutations are non-synonymous mutations in the candidate resistance loci.

5. Detected top candidate genes can be analyzed further with pathway enrichment analysis. It might reveal some potential drug resistance pathways.

Thank you very much for your insightful comment. We agree, pathway enrichment analysis would be an excellent addition to our analysis. We did initially perform this analysis but found that no pathways or gene ontology terms were specifically enriched in the Indonesian population, nor in the pre-2014 vs. post-2014 populations, so unfortunately could not provide any further insight here.

6. It would be much clearer if a “Locuszoom” figure that enlarges the significantly associated regions, such as chr10:320-330kb, displayed the position of candidate variants or genes in detail was added.

We agree that for the readers, it would be clearer to expand on the critical loci in chromosome 10 (320-330kb) that has signals in IBD, *iHS*, *Rsb*, and *XP-EHH* analyses. We have updated our Supplementary figure (S9) which shows the IBD on chromosome 10, and we have highlighted the position of *pvm_{dr}1* in relation to chr10:320-330kb. We have also added an additional supplementary figure (S10), which contains a locus zoom of this region of interest.

7. The signals were detected mainly due to the variant allele frequency differences. Drugs can be strong selection pressures, and other factors such as human and vector genomic background and climate changes during these years can also lead to the parasite gene variant allele frequency changes. These points should be included in the discussion part. **We agree that this context is important. We have addressed this in our Discussion (L427-435): “Allele frequency trajectories are not solely driven by selection pressures exerted by drug regimes, as changes in the human or vector genomic backgrounds and climate change may also be at play. However, the increasing frequency of novel *pvm_{rp}1* SNPs that has occurred after introduction of dihydroartemesinin-piperaquine in the Indonesian population in addition to continued selection at the upstream *pvm_{dr}1* locus is consistent with the hypothesis that adaptation of parasites to changes in drug regimens is polygenic in nature, as seen in *P. falciparum* with CQR^{17,18}. The appearance of novel *pvm_{rp}1* haplotypes under the changed drug regime could be to mitigate the cost of previously fixed alleles, such as those in PvMDR1, or could in fact be resistance-conferring.”**

REVIEWER 3

Reviewer #3: (Remarks to the Author):

This study makes an important contribution to our understanding of the genomics and evolution of *Plasmodium vivax* malaria. It uses an array of population genetics tools to explore the evidence for recent selection occurring in *P. vivax* populations and focuses on potential cases of selection for drug resistance. The rich dataset and the range of techniques employed in the analysis are strengths of this work. I believe that the importance of the question and the scale of the work make it appropriate for publication in *Nature Communications*. However, the manuscript would benefit from revisions to improve clarity, and I have some questions about how some of the analysis was done (see below).

We thank Reviewer #3 for their valued comments on the important contribution this study will make to *P. vivax* malaria research, and their recognition of the extensive dataset and range of techniques we used to answer important questions about the evolution of drug resistance in *P. vivax*.

1. Readers of the manuscript would benefit from a clear description of historical and current antimalarial drug use in the regions where the isolates came from, and a summary of what is known about drug resistance in *P. vivax* in these different geographical areas. The manuscript already provides some of this sort of information, e.g. “Indonesian Papua isolates have maintained a CQR phenotype despite the removal of drug pressure exerted by chloroquine since its contraindication in 2004”; “Cambodia and Vietnam...both countries with declining chloroquine sensitivity”. This information is largely concentrated in the Discussion. Having this information summarised in a table (e.g. a table with headings: Country, Number of isolates in this analysis, Antimalarial use history, Evidence for *P. vivax* antimalarial resistance within this country) would be extremely useful for the reader to help them interpret the results (and if some of the information is unavailable that is also useful for the reader to know). The designation of countries as having “low grade” or “high grade” resistance could also be explained in this table, because this definition is unclear in the manuscript.

The suggested table is a great idea and synergises with comments from other reviewers. We now present supporting supplementary tables (S2 and S3) with all this information.

2. The title includes the phrase “genome wide association study”, which implies a comparison between groups of known properties (e.g. drug resistant and non drug resistant isolates). I suggest not using the phrase “genome wide association study” at all because the actual drug resistance profiles of these specific isolates are not part of this analysis. **We agree, and have changed the manuscript title to “Genomic analysis of global *Plasmodium vivax* populations reveals insights into the evolution of drug resistance”**

3. The abstract states “Differential selection metrics applied between isolates from low-grade and high-grade CQR regions revealed sweeps in a locus proximal to *pvm*dr1 and in transcriptional regulation genes”. This implies that low grade and high grade CQR regions were specifically compared, but as far as I can tell from the results, all possible comparisons were made between countries and regions (using the *Rsb* metric), and some of the interesting results which emerged showed evidence for differentiation of potential resistance loci between certain “low grade” CQR regions and Indonesia (figure 2). It took me a while to realise this is what the abstract meant so it may be worth re writing this part of the abstract to make it clearer.

We agree, and we have modified our Abstract accordingly (L27-L38).

4. The Methods do not make clear how statistical analysis of the results were carried out. For example, certain IBD values are referred to as being significant (lines 176, 177, 187). How was this significance determined? Figures 2 and 4 indicate cutoffs for significance – what is the justification for using these particular cutoffs?

Thanks for noticing this. We have removed the use of “significant” because this method identifies critical loci, which we discussed in our Methods (L557-L558): “critical loci were identified using 10 kb sliding windows which included at least 5 SNPs with a P value of $< 1 \times 10^{-4}$ for *iHS* and $< 1 \times 10^{-5}$ for *Rsb* and *XPEHH*. These cut-offs have been used previously in prior studies (DOI: <https://doi.org/10.1038/s41467-021-23422-3>, DOI: <https://doi.org/10.1016/j.lana.2022.100420>), and have been calculated using a Gaussian approximation method from this study here (DOI: <https://doi.org/10.1371/journal.pgen.1005131>). We have explained this for more clarity in

our Methods section on L558-559: “These cutoffs were calculated using a Gaussian approximation method”

5. The Methods refer to in-house scripts for the IBD analysis (line 501). It would be good to indicate to readers how these could be obtained.

The in-house scripts are available on our GitHub page, and the link is provided in our Methods on L564-L565. We have also provided a code availability statement on L584: “All scripts use open-source software (see “Methods”).”

5. *pvmrp1* is described in the abstract as “as an emerging candidate for piperazine resistance”, but in line 324 as “an emerging putative marker for primaquine resistance”. I am pretty sure, given the rest of the discussion, that line 324 is a typo, however it does raise an interesting point. Evidence from *P. falciparum* (reference [30] in the manuscript) would suggest that mutations in *pvmrp1* could be involved in resistance to either piperazine or primaquine. Is the IBD pattern the authors report for *pvmrp1* likely to be a consequence of selection from piperazine or primaquine, or both? The likelihood of either drug exerting selection pressure in different geographical regions needs to be discussed more in the manuscript, so that the case for *pvmrp1* as a marker of resistance is made more clearly.

Thank you for the insightful comment. In our analysis dataset, primaquine is used in every single country for radical cure. In Indonesian Papua, the current antimalarial, dihydroartemisinin-piperazine + primaquine) superseded the chloroquine + primaquine strategy in 2004. We only observe these novel *pvmrp1* SNPs and F_{WS} patterns with novel drug pressure from dihydroartemisinin-piperazine, and therefore hypothesise that this signal is likely to be a consequence of selection from piperazine, which agrees with an *ex vivo* susceptibility study that found decreasing susceptibility of Indonesian Papua isolates to piperazine (doi: 10.1016/j.ijpddr.2021.06.002).

We have made the case for this more clearly in the manuscript, correcting a typo in line 324 (now 340), and improved our Discussion:

- **L338-L341: “Additionally, our selection and temporal SNP dynamic analyses suggest that *pvmrp1* is a promising molecular research and surveillance target, potentially linked to piperazine resistance. Further investigation is required to fully understand its contribution to antimalarial resistance in *P. vivax*.”**
- **L416-L423: “Dihydroartemisinin-piperazine has replaced chloroquine as the frontline antimalarial against *P. vivax* in Indonesia, with primaquine still remaining in the treatment strategy to target hypnozoite stages. We observed that, since its introduction, increasing dihydroartemisinin-piperazine pressure on Indonesian Papuan isolates over time is correlated with an increasing prevalence of these *pvmrp1* SNPs. Between 2008-2018, *ex vivo* susceptibility of *P. vivax* to piperazine in the Indonesian Papua population has declined²⁹. Although clinical efficacy of dihydroartemisinin-piperazine is reportedly high in Indonesian Papua, therapeutic efficacy studies have shown day 42 recurrence rates ranging from 1.2%⁵⁰ to 11.3%⁵¹.”**
- **L429-435: “However, the increasing frequency of novel *pvmrp1* SNPs that has occurred after introduction of dihydroartemisinin-piperazine in the Indonesian population in addition to continued selection at the upstream *pvmdr1* locus is consistent with the hypothesis that adaptation of parasites to changes in drug regimens is polygenic in nature, as seen in *P. falciparum* and CQR^{17,18}. The appearance of novel *pvmrp1* haplotypes under the changed drug regime could be to mitigate the cost of previously fixed alleles or could in fact be resistance-conferring.”**

The following small points may require correction or alteration to improve clarity. > I found it hard to distinguish the single West African isolate in panels A and B of figure 1. A different colour

scheme would be worth considering, so that there is a bigger contrast between the colour for East Africa and the colour for West Africa.

We have modified the colour for the West African isolate in Figure 1 accordingly.

There are some contradictions between the text and the supplementary tables in terms of whether they refer to the top 5% or the top 1% of the relevant results. E.g. line 156 refers to the top 5% of IBD results but table S5 says top 1%; line 196 refers to the top 1% of genes under recent positive selection measured using the iHS metric, but table S7 says top 5%. **Thank you for noticing this, and we apologise for the confusion. For the IBD results, we have added an additional supplementary table (S8) showcasing the top 5% results too. For the iHS metric, Table S7 had a typo, and does indeed show the top 1% of results.**

Line 392 of the discussion refers to genes involved in a selective sweep on chromosome 14 in the Indonesian Papua isolates. This part of the discussion suggests the genes include CLAG, but CLAG is not mentioned in the results relating to this sweep on chromosome 14 (lines 300 and 301). **Thank you, this has been included now on line 316.**

Line 359 refers to the Y976F mutation. It would be clearer to insert “in *pvmdr1*” to remind readers about the gene in which the mutation occurs.

Thank you very much, this has now been implemented on line 376: “While in vitro susceptibility studies have shown a role of the Y976F mutation of *PvMDR1* in modulating CQR in isolates from Indonesian Papua...”

The paragraph starting at line 372 was hard for me to follow. Reorganizing the order of the paragraph may make things easier, e.g. starting with the sentence “Considering the trend of declining chloroquine sensitivity [...] we hypothesised that there should be signals of selection in *pvmdr1* ...”, then going on to discuss the evidence that was found.

Thank you very much, we have modified our language accordingly for clarity on lines 3910-392: “Given the trend of declining chloroquine sensitivity in regions such as Sabah, Malaysia⁴¹, Vietnam⁴⁵, and China-Myanmar border⁴⁰, we hypothesised that if *pvmdr1* is mediating CQR, there should be detectable signals of selection in *pvmdr1* indicative of this resistance.”

REVIEWERS' COMMENTS

Reviewer #1 (Remarks to the Author):

I accept the responses in the revision and have no further comments.

Thank you very much for the time taken to review our manuscript.

Line 120-124. Fig.1a and 1b. 1b appears before 1a in the text. They can be switched to have 1a appear first.

Now corrected.

Nature portfolio

Competing interests: Both boxes are checked?

We declare no competing interests and have updated our Reporting Summary file.

Reviewer #2 (Remarks to the Author):

We thank Reviewer 2 for their time and comments on the work we presented here, and for their support of an ECR in co-reviewing our manuscript.

Reviewer #3 (Remarks to the Author):

Thank you to the authors for their revised manuscript. The revisions make their results much clearer, and the new table S2 is a particularly useful addition. There are two things which I think still need to be clarified before the manuscript is published (points 1 and 2 below), and I also have some additional small comments that I hope will be helpful.

1. The section of the results which describes the IBD analysis is still unclear. I asked about whether the top 1% or top 5% of IBD values were analysed, and the authors have provided tables of both 1% and 5%. However it is still not clear how these are to be used or interpreted by the reader. The text (line 155 onwards) mentions 5% but then refers to table S7 (the top 1% table). It would be clearer to present the results in a way that says something like "genomic regions with consistently high IBD fractions (i.e. >?% in ? countries) included [xyz]". I also wanted to query whether the sentence "we focused on windows with the top 5% of mean IBD fractions, within or around drug resistance loci" is necessary at all? That sentence implies that the authors only considered windows that were already close to drug resistance loci, and yet the rest of the paragraph suggests that they considered all regions, and then found that some regions with the highest IBD values were associated with drug resistance loci (line 162). If my understanding is correct (i.e. they looked at all regions, not just regions already close to drug resistance loci), then lines 155-158 need to be re written for clarity. **Thank you very much for your insight here. We agree that interpreting both the 1% (S7) and 5% tables (S8) would be confusing to the reader. For clarity purposes, we have decided to remove the 5% table (S8).**

To your second point, we considered all regions, and not just those already proximal to drug resistance loci. To be more explicit, we have rephrased this paragraph on lines 155-158: "To investigate patterns of shared ancestry intrachromosomally, we analysed genome-wide IBD fractions calculated across 10 kb sliding windows, investigating specifically genomic regions falling in the top 1% of fractions" (Fig. S6). Due to the high genetic relatedness of isolates from Malaysia, Mexico, and Panama..."

2. I asked about the cross population analysis in my earlier review. The authors have now updated the abstract so that it is much clearer- thank you very much for this. Now that the abstract is clear, the section of the text that describes the cross population Rsb analysis (line 212 onwards) is a bit unclear (and in fact, I think the extra sentence added in line 213 makes it less clear!) As I understand it, the

authors made all possible pairwise comparisons between countries and *then* interpreted their results in terms of high grade and low grade CQR. The new addition in line 213 implies that the authors already chose the comparisons to make, due to the differences in CQR (but it that was the case, why do they only talk about Indonesia in this section, and not Papua New Guinea which also has high grade CQR?) I think that the logical progression of the ideas in section starting at line 212 should be something like: (i) cross population Rsb analysis was carried out; (ii) Indonesia looked particularly different from other countries in this analysis, then followed by the discussion/potential explanation point that Indonesia happens to have especially high grade CQR. I may have misunderstood, so please re write this section to reflect what the authors want to say, but I wanted to explain why I am still struggling with the way this section has been written.

- **Thank you for your comments here. We focused specifically on the Indonesian Papua population because the number of monoclonal isolates from Papua New Guinea (N = 15), in comparison to Indonesian Papua (N = 106) is limited (selection analysis is performed on monoclonal isolates only).**
- **Moreover, Papua New Guinea is a relatively understudied country in terms of therapeutic efficacy studies (both for previous use of chloroquine and their current front-line antimalarial), which again, are limited in number (detailed in our S2 table).**
- **We believed it necessary to choose populations that have a robust enough sample size in addition to adequate, recent, therapeutic efficacy studies or *ex vivo* susceptibility studies to be able to draw sound and substantiated conclusions from this analysis.**
- **We completely agree that this section needs to be rewritten for clarity, which we have done so on lines 211-214: “To identify signals of directional selection, the cross-population metric, *Rsb*, was used at both a country and regional level, specifically comparing Indonesia with other countries due to known differences in CQR status. Although PNG is also a region of high-grade CQR, due to limited sample size, we excluded it from cross-population selection analysis.”**
- **In our original submission, we had also excluded PNG from further investigations in IBD analyses, which we stated in our discussion on lines 400-402.**

Small suggestions:

3. The new table S2 is excellent and could be referred to in more places, e.g. line 60 of the current manuscript (currently it looks as though that paragraph needs more referencing - directing the reader to table S2 will help). I could not find China in the table, even though I believe there were >10 samples from China (Table S1). This may simply be an omission and China can be added, but if there is a reason China is not included, this should be explained.

Thank you for the positive comment on the new Table S2. We would prefer not to reference it in the Introduction as it is a result. We have now added multiple studies used in generating Table S2 on line 60. China has >10 isolates, but as we stated on line 498 in our methods, “Finally, we excluded studies from countries within our dataset with fewer than 10 publicly available *P. vivax* genome sequences or those certified malaria-free by the WHO”, of which the latter was the case for China in 2021. We have added this in the title for Table S2 for additional clarity and appreciate your observation. “S2: ... Countries now certified malaria-free have been excluded.”

4. The sentence "We therefore evaluated the prevalence of resistance haplotypes (non synonymous mutations) in all isolates..." (lines 230-231) is a bit confusing because the phrase "resistance haplotypes" implies a haplotype that has definitely been associated with resistance, yet the rest of the section refers to non-synonymous mutations, some of which have some evidence for an association with resistance and some of which do not (or do not yet!) If I have understood correctly, then it would be clearer to say something like "We therefore evaluated the prevalence of non-synonymous mutations affecting known or candidate antimalarial resistance loci". The subheading (line 228) could also be slightly altered to say "putative" drug resistance mutations (or similar).

We agree completely to make it explicit that these mutations are putatively associated with resistance. We have changed lines 230-231, which are now 231-232, stating: **“We therefore evaluated the prevalence of non-synonymous mutations in all isolates in genes with a putative association to antimalarial resistance in *P. vivax*.”**

The subheading has also been amended to: **“Sub-regional differences in the frequency of putative resistance mutations in *P. vivax* populations”**

5. In line 368, I believe the word "sites" or possibly "residues" is missing, i.e. "structurally corresponded with sites/residues linked to... "

Thank you. We have now added the word “residues”.

6. Line 431 refers to "continued selection at the upstream *pvm*dr1 locus" in Indonesian Papua. What is the evidence for continued selection? I was also not clear what the "upstream" region was, since I thought that the region with the potential selective sweep was downstream of *pvm*dr1.

The potential sweep is indeed downstream *pvm*dr1. We agree that this section of the paragraph was not particularly clear, so have modified it to remove this sentence as there was some repetition.

It is now on lines 432-437: “Allele frequency trajectories ... may also play a role. The appearance of novel *pvm*rp1 haplotypes after the introduction of dihydroartemisinin-piperazine in Indonesia could reflect changing local dynamics, could mitigate the cost of previously fixed alleles, such as those in *pvm*dr1, or could in fact be resistance-conferring.”

7. In lines 403 to 404 the authors hypothesise that the region where they have identified a putative selective sweep may contain "neutral alleles". I was not sure what was being argued here. I agree that if there was a temporary selective pressure between 2012 and 2015 it may not have been chloroquine, and I also agree that it's possible that "changing local dynamics" are responsible for the pattern, but I think it needs to be made clearer if the authors are saying that, in fact, the pattern they have found may not indicate any form of selection (i.e. all the variation in the region is neutral), or that they think it still does indicate some form of selective sweep, but not necessarily one caused by antimalarial drugs.

To make our position on this explicit, we have modified the phrasing of our argument on lines 404-411 to the following: “Given the temporally transient nature of this genomic signature, several hypotheses can be generated. Firstly, one could hypothesise that these SNPs have no impact on the parasite’s fitness or ability to withstand chloroquine pressure and are therefore neutral alleles not linked to CQR. Conversely, one could hypothesise that these SNPs reflect changing local dynamics, considering that chloroquine is still readily available in the private sector in Indonesian Papua⁴⁷. In this scenario, the temporal signature could be a result of indirect selection, as the SNPs may be beneficial in periods of increasing CQ use, but non-beneficial in periods of increasing DHA-PPQ use.”